# RegionE: Adaptive Region-Aware Generation for Efficient Image Editing

**Pengtao Chen**[1]   **Xianfang Zeng**[2‡]   **Maosen Zhao**[1]   **Mingzhu Shen**[3]   **Peng Ye**[1]
**Bangyin Xiang**[4]   **Zhibo Wang**[2]   **Wei Cheng**[2]   **Gang Yu**[2]   **Tao Chen**[1,5*]
[1] Fudan University    [2] StepFun    [3] Imperial College London
[4] Communication University of China    [5] Shanghai Innovation Institute
**Code:** https://github.com/Peyton-Chen/RegionE

## Abstract

Recently, instruction-based image editing (IIE) has received widespread attention. In practice, IIE often modifies only specific regions of an image, while the remaining areas largely remain unchanged. Although these two types of regions differ significantly in generation difficulty and computational redundancy, existing IIE models do not account for this distinction, instead applying a uniform generation process across the entire image. This motivates us to propose **RegionE**, an adaptive, region-aware generation framework that accelerates IIE tasks without additional training. Specifically, the RegionE framework consists of three main components: **1) Adaptive Region Partition**. We observed that the trajectory of unedited regions is straight, allowing for multi-step denoised predictions to be inferred in a single step. Therefore, in the early denoising stages, we partition the image into edited and unedited regions based on the difference between the final estimated result and the reference image. **2) Region-Aware Generation**. After distinguishing the regions, we replace multi-step denoising with one-step prediction for unedited areas. For edited regions, the trajectory is curved, requiring local iterative denoising. To improve the efficiency and quality of local iterative generation, we propose the Region-Instruction KV Cache, which reduces computational cost while incorporating global information. **3) Adaptive Velocity Decay Cache**. Observing that adjacent timesteps in edited regions exhibit strong velocity similarity, we further propose an adaptive velocity decay cache to accelerate the local denoising process. We applied RegionE to state-of-the-art IIE base models, including Step1X-Edit, FLUX.1 Kontext, and Qwen-Image-Edit. RegionE achieved acceleration factors of 2.57×, 2.41×, and 2.06×, respectively, with minimal quality loss (PSNR: 30.520–32.133). Evaluations by GPT-4o also confirmed that semantic and perceptual fidelity were well preserved.

## 1 Introduction

In recent years, diffusion models (Rombach et al., 2022) have achieved rapid progress in generative tasks, particularly in visual generation, where state-of-the-art models can synthesize highly realistic images. Within this context, the task of editing existing images according to user requirements has gradually emerged as an important direction (Kawar et al., 2023). Recently, diffusion-based foundation models, such as FLUX.1 Kontext (Labs et al., 2025), Qwen-Image-Edit (Wu et al., 2025), and Step1X-Edit (Liu et al., 2025b; Yin et al., 2025), have been developed. These models can perform precise image editing using only textual instructions, offering a novel solution for instruction-based image editing and providing more powerful tools for image post-processing (Choi et al., 2024).

Although diffusion-based IIE models can achieve impressive editing results, their high inference latency limits their use in real-time applications. Previous research on efficient diffusion inference has primarily focused on image generation. For instance, some studies reduce model parameters through pruning (Castells et al., 2024), others decrease model bit-width via quantization (Zhao et al., 2025a), and some accelerate attention (Zhang et al., 2025; Chen et al., 2025), and some employ distillation

---

*Corresponding author. ‡Project leader. Work was done when interned at StepFun.

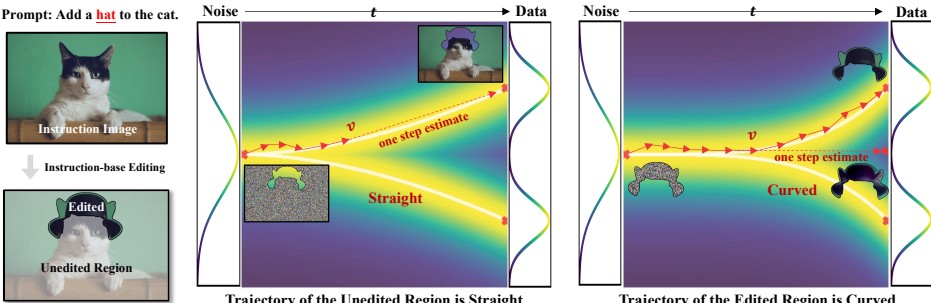

Figure 1: Trajectories of different regions in the IIE task. In unedited regions, the trajectory is nearly linear, allowing early-stage velocity to provide a reliable estimate of the multi-step denoised images, including the final result. In contrast, edited regions exhibit curved trajectories, making the final image harder to predict. Despite this, the velocity between consecutive timesteps remains consistent.

to reduce model size (Kim et al., 2023) and the number of timesteps (Sauer et al., 2024). In the two-stage inversion-based editing paradigm (Pan et al., 2023; Wang et al., 2025), redundancy in the inversion and denoising stages has been analyzed, leading to methods like EEdit (Yan et al., 2025) that accelerate both stages simultaneously. However, for the emerging denoising-only paradigm of IIE, the redundancy and feasibility of efficient inference remain largely unexplored.

Our study reveals that current IIE models exhibit two significant types of redundancy: 1) Spatial Generation Redundancy. Unlike image generation tasks, which require reconstructing the entire image, IIE models often need to modify only local regions specified by the instructions, while the remaining areas remain essentially unchanged. For example, as shown in Figure 1, the model edits only the region around the hat. Nevertheless, IIE models apply the same computational effort to both edited and unedited areas, resulting in significant redundancy in the latter. 2) Redundancy across diffusion timesteps. First, at neighboring timesteps, the key and value within the attention layers at the same network depth are highly similar. Second, in the middle stages of denoising, the velocity output by the diffusion transformer (DiT) at adjacent timesteps is also highly similar.

To mitigate spatial and temporal redundancy in IIE models, this paper introduces RegionE, a training-free, adaptive, and region-aware generative framework that accelerates the current IIE models. Firstly, we observed that the trajectories of edited regions are often more curved, making it difficult to accurately predict the final edited results at early timesteps, as shown in Figure 1. In contrast, unedited regions follow nearly linear trajectories, allowing more reliable predictions from the same early steps. Based on this observation, RegionE introduces an Adaptive Region Partition (ARP), which performs a one-step estimation for the final image in the early stage and compares its similarity with the reference (instruction) image. Regions with high similarity (minimal change after editing) are classified as unedited, whereas regions with low similarity are classified as edited. Then, we perform region-aware generation on the two separated parts. Specifically, We replace multi-step denoising with one-step estimation for the unedited areas and apply region-iterative denoising for edited areas. During edited region generation, RegionE discards unedited region tokens and instruction image tokens, and effectively reinjects global context into local generation through our proposed Region-Instruction KV Cache (RIKVCache), which leverages the similarity of key and value across timesteps. This process primarily addresses redundancy in spatial. Finally, regarding temporal redundancy, we find that the velocity outputs of DiT at adjacent timesteps are highly consistent in direction but decay in magnitude over time, with the decay dependent on the timestep. To exploit this property, RegionE introduces an Adaptive Velocity Decay Cache (AVDCache), which accurately models this pattern and further accelerates the region generation process. Experimental results demonstrate that RegionE achieves speedups of approximately 2.57×, 2.41×, and 2.06× on Step1X-Edit, FLUX.1 Kontext, and Qwen-Image-Edit, respectively, while maintaining PSNR values of 30.520, 32.133, and 31.115 before and after acceleration. Evaluations using GPT-4o further indicate that the perceptual differences are negligible, confirming that RegionE effectively eliminates redundancy in IIE tasks without compromising image quality.

The contributions of our paper are as follows:

- We observe that in IIE tasks, unedited regions exhibit nearly linear generation trajectories, allowing early-stage velocities to provide reliable estimates for multi-step denoised images, including the final image. In contrast, edited regions follow more curved trajectories, making the final image harder to predict. Nevertheless, the velocity remains consistent across consecutive timesteps.

- We propose RegionE, a training-free, efficient IIE method with adaptive, region-aware generation. It reduces spatial redundancy by performing early adaptive predictions for edited and unedited regions and generating each region locally in subsequent stages, while mitigating temporal redundancy via a velocity-decay cache across timesteps.

- RegionE achieves 2.57×, 2.41×, and 2.06× end-to-end speedups on Step1X-Edit, FLUX.1 Kontext, and Qwen-Image-Edit, while maintaining PSNR (30.520, 32.133, 31.115) and SSIM (0.939, 0.917, 0.937). Evaluations with GPT-4o further confirm that no quality degradation occurs.

## 2 RELATED WORK

**Efficient Diffusion Model.** Although few efficient methods have been developed specifically for IIE models, a variety of acceleration techniques have been proposed for diffusion models more generally. From the perspective of parameter redundancy, researchers have introduced pruning methods such as Diff-Pruning Fang et al. (2023) and LD-Pruner (Castells et al., 2024), quantization methods such as PTQ4DM (Shang et al., 2023) and SVDQuant (Li et al., 2024a), distillation methods such as BK-SDM (Kim et al., 2023) and CLEAR (Liu et al., 2024), and early-stopping strategies such as ES-DDPM (Lyu et al., 2022). From the perspective of temporal redundancy, methods like DeepCache (Ma et al., 2024), $\Delta$-DiT (Chen et al.), FORA (Selvaraju et al., 2024), MD-DiT (Shen et al., 2024) and TeaCache (Liu et al., 2025a) reuse intermediate features across timesteps, while approaches such as LCM (Luo et al., 2023) and ADD (Sauer et al., 2024) reduce the number of timesteps through model distillation. From the perspective of spatial redundancy, RAS (Liu et al., 2025c) observes that at each diffusion timestep, the model may focus only on semantically coherent regions; therefore, only those regions need to be updated, thereby accelerating image generation. Similarly, ToCa (Zou et al., 2024a) and DuCa (Zou et al., 2024b) note that during denoising, different tokens exhibit varying sensitivities, and dynamically updating only a subset of tokens at each timestep can further accelerate image generation (Kong et al., 2025). In contrast to the methods above, RegionE leverages the trajectory characteristics unique to IIE tasks, while simultaneously addressing both spatial and temporal redundancies in diffusion-based image editing to achieve accelerated generation.

**Image Editing.** Image editing is an essential task in the field of generative modeling. In the early U-Net (Ronneberger et al., 2015) era, ControlNet (Zhang et al., 2023b) introduced a robust editing solution through a repeat-structure design, and InstructPix2Pix (Brooks et al., 2023) introduced an editing method that involves channel expansion. As research advanced, inversion-based methods (Pan et al., 2023; Wang et al., 2025) gradually became the dominant approach. These methods apply noise to the original image in the latent space and then recover the edited result through a denoising process. However, this paradigm involves both inversion and denoising stages, which increases complexity. At the same time, IIE models began to emerge. Approaches such as InstructEdit (Wang et al., 2023), MagicBrush (Zhang et al., 2023a), and BrushEdit (Li et al., 2024b) employed modular pipelines, in which large language models generate prompts, spatial cues, or synthetic instruction–image pairs to guide diffusion-based editing. Most of these approaches, however, are task-specific and lack generality. More recently, a new class of IIE has been developed to improve general-purpose editing. These models rely solely on textual instructions, without requiring masks or task-specific designs, and still achieve effective editing performance. Concretely, they leverage MLLMs or advanced text encoders to provide richer semantic control signals, and feed both the target image and noise into a DiT (Peebles & Xie, 2023) architecture to enhance image alignment. In this work, we propose an adaptive, region-aware acceleration method for these emerging IIE models. Although prior work has explored local editing, these studies primarily aim to enhance editing capability rather than improve efficiency. Moreover, methods such as (Simsar et al., 2024) and (Guo & Lin, 2023) follow the InstructPix2Pix paradigm, while (Mo et al., 2024; Couairon et al., 2022; Avrahami et al., 2022; Yang et al., 2024) operate within inversion-based or mask-dependent editing frameworks. In contrast, we investigate the problem under the emerging MLLM-assisted IIE paradigm and, for the first time, identify an early-stage region-partitioning strategy in modern flow-matching models that enables an effective acceleration mechanism.

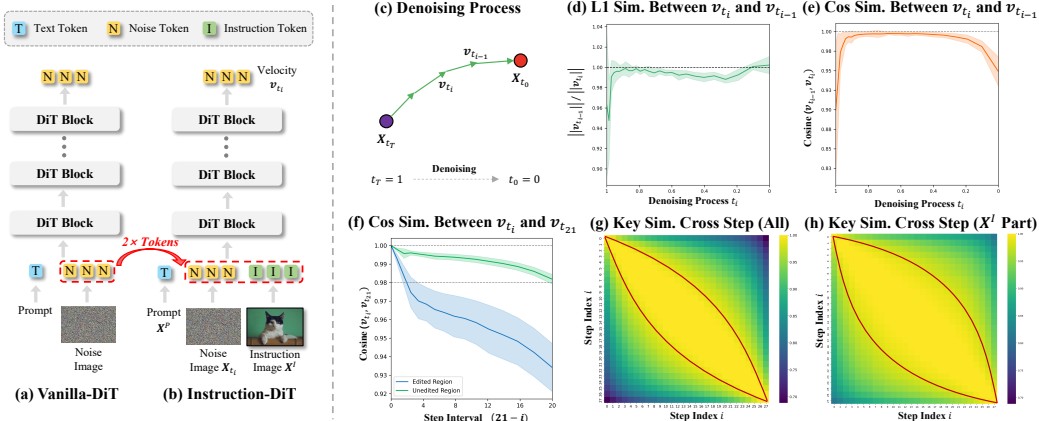

Figure 2: Comparison between traditional DiT and DiT in IIE (a, b). Symbolic visualization of the denoising process (c). L1 and cosine similarities of velocities between adjacent timesteps during denoising (d, e). Cosine similarity between velocities after $t_{21}$ in edited and unedited regions with $v_{21}$ (f). Cross-step key similarity (g) and cross-step similarity of instruction-related keys (h).

## 3 PRELIMINARY

**Flow Matching & Rectified Flow.** Flow matching (Lipman et al., 2022) has become a widely adopted training technique in advanced diffusion models. It facilitates the transfer from a source distribution $\pi_1$ to a target distribution $\pi_0$ by learning a time-dependent velocity field $v(x, t)$. This velocity field is used to construct the flow through the ordinary differential equation (ODE):

$$\frac{d\phi_t(x)}{dt} = v(\phi_t(x), t), \phi_1(x) \sim \pi_1. \tag{1}$$

Rectified Flow (Liu et al., 2022) simplifies this process through a linear assumption. Given that $X_1$ follows a noise distribution $\pi_1$ and $X_0$ follows the target image distribution $\pi_0$, the equation is

$$X_t = (1 - t)X_0 + tX_1, t \in [0, 1]. \tag{2}$$

Differentiating both ends with respect to timestep $t$ yields: $\frac{dX_t}{dt} = X_1 - X_0$. The velocity of the rectified flow $v(X_t, t)$, always points in the direction of $X_1 - X_0$. Therefore, the training loss is minimized by reducing the deviation between the velocity and $X_1 - X_0$:

$$\mathcal{L} = \mathbb{E}_t\left[||(X_1 - X_0) - v(X_t, t)||^2\right]. \tag{3}$$

The inference process involves starting from $X_1$ and iteratively solving for $X_0$ in reverse, using the learned velocity $v(X_t, t)$. In practice, we typically use a discrete Euler sampler, which discretizes the timestep $t_i(i \in \mathbb{N}^T, t_T = 1, t_0 = 0)$ and approximates:

$$X_{t_{i-1}} = X_{t_i} - \Delta t_{i,i-1} \cdot v(X_{t_i}, t_i), \Delta t_{i,i-1} = t_i - t_{i-1}. \tag{4}$$

After $T$ iterations, the final target image $X_0$ is obtained. This paper, therefore, targets the IIE task and optimizes the inference process of $T$ iterations in Equation 4.

**Instruction-Based Editing Model.** Recent IIE models, such as Step1X-Edit (Liu et al., 2025b), FLUX.1 Kontext (Labs et al., 2025) and Qwen-Image (Wu et al., 2025), follow the same paradigm, as shown in Figure 2b. In these models, the velocity field is estimated using Instruction-DiT, the variants of DiT (Peebles & Xie, 2023). The input to Instruction-DiT consists of three types of tokens: text (prompt) tokens $X^P$, noise tokens $X_{t_i}$, and instruction tokens $X^I$. The noise token corresponds to the generation of the target image, while the text token carries the instruction information. The instruction token is specific to the editing task, representing the part of the image to be edited. Notably, the counts of noise and instruction tokens are roughly comparable and substantially higher than that of text tokens. Temporally, the text and instruction tokens serve as static control signals throughout the denoising process, whereas the noise token evolves dynamically at each timestep. Since Instruction-DiT is designed to predict only the noise component, the model's output corresponds exclusively to the portion represented by the noise token. To simplify the expression, the Instruction-DiT mentioned below will be referred to simply as DiT.

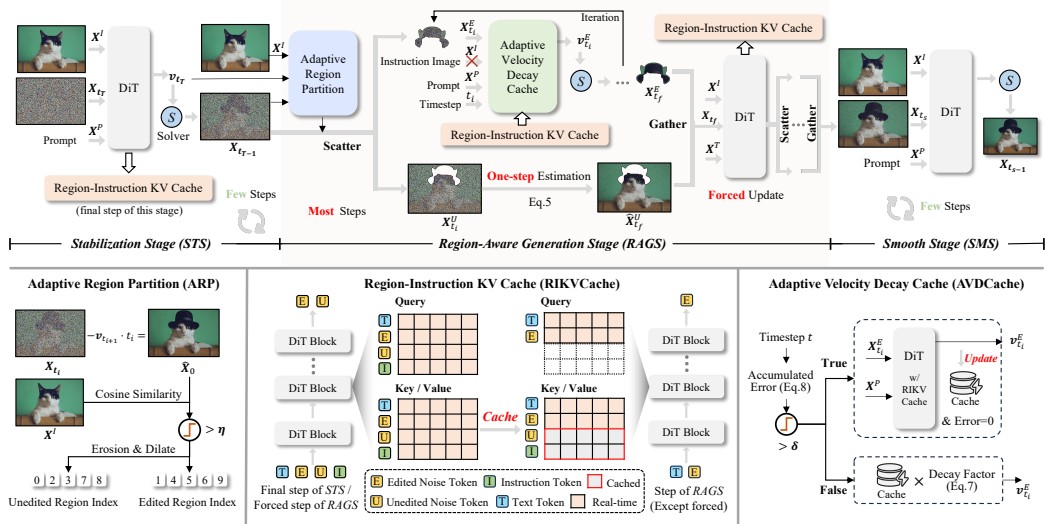

Figure 3: **Overview of the RegionE**. RegionE consists of three stages: STS, RAGS, and SMS. In the STS, no acceleration is applied due to unstable DiT outputs, and all KV values are cached at the final step. In the RAGS, an Adaptive Region Partition distinguishes between edited and unedited regions: unedited regions are denoised in one step, while edited regions are generated iteratively. This iterative generation process leverages RIKVCache for injecting global information and AVDCache for acceleration. Certain forced-update steps aggregate the full image to refresh RIKVCache with complete DiT computation. Finally, in the SMS, several full denoising steps are performed to eliminate artifacts along the boundaries between edited and unedited regions.

## 4 METHODOLOGY

This section introduces RegionE, a method that accelerates the IIE model without additional training. The workflow is shown in Figure 3. RegionE consists of three stages: the Stabilization Stage (STS), the Region-Aware Generation Stage (RAGS), and the Smooth Stage (SMS).

**Stabilization Stage.** In the early steps of denoising, the input $\boldsymbol{X}_{t_i}$ to DiT is close to Gaussian noise (i.e., the signal-to-noise ratio is low). This leads to oscillations in DiT's velocity estimation (see Figure 2d and 2e). Since the estimates at this stage are inherently unstable, it is not suitable for acceleration. Therefore, we keep the original sampling process unchanged. Additionally, at the last step of this stage, we save the Key and Value in each attention layer of DiT, denoted as $\boldsymbol{K}^C$ and $\boldsymbol{V}^C$.

**Region-Aware Generation Stage.** This stage is the core component of RegionE and consists of three parts: adaptive region partition, region-aware generation, and adaptive velocity decay cache. The first two parts primarily address spatial redundancy in IIE, while the third further reduces temporal redundancy across timesteps.

**Adaptive Region Partition.** After the stabilization stage, the output of DiT becomes stable. As previously observed, the generation trajectories in the edited regions are curved, whereas those in the unedited regions are straight, as shown in Figure 1 and 2f. Therefore, for the unedited regions $\boldsymbol{X}_{t_i}^U$, we can accurately estimate $\hat{\boldsymbol{X}}_{t_f}^U$ at any timestep $t_f(f < i)$ using one-step estimation:

$$\hat{\boldsymbol{X}}_{t_f}^U = \boldsymbol{X}_{t_i}^U - \boldsymbol{v}^U(\boldsymbol{X}_{t_i}^U, t_i) \cdot \Delta t_{i,f}. \tag{5}$$

When $t_f = 0$, this corresponds to estimating the final unedited regions $\hat{\boldsymbol{X}}_0^U$, which is nearly identical to the true $\boldsymbol{X}_0^U$. However, using Equation 5 for the edited region does not accurately estimate $\hat{\boldsymbol{X}}_0^E$. Based on this difference between the edited and unedited regions, we propose an adaptive region partition (ARP), as illustrated in the lower-left corner of Figure 3. Given the velocity $\boldsymbol{v}_{t_{i+1}}$ at the beginning of the region-aware generation stage and the noisy image $\boldsymbol{X}_{t_i}$, the final edited result $\hat{\boldsymbol{X}}_0$ can be estimated in one step using Equation 5. This estimate is reliable in unedited regions but less accurate in edited ones. Since the unedited region undergoes minimal change before and after editing,

we can compute the cosine similarity between the estimated image $\hat{X}_0$ and the instruction image $X^I$ along the token dimension. Regions with sufficiently high similarity ($>$ threshold $\eta$), that is, small changes before and after editing, are considered unedited regions, while the remainder is treated as the edited region. To account for potential segmentation noise, morphological opening and closing operations are applied to make the two regions more continuous and accurate.

**Region-Aware Generation.** After identifying the edited and unedited regions, we apply Equation 5 to the unedited region to directly estimate the denoised image $X_{t_f}^U$ at the next timestep $t_f$ in one step, thereby saving computation for the unedited region. For the edited region, our implementation is as follows: first, the input to DiT is changed from $[\boldsymbol{X}^P, \boldsymbol{X}_{t_i}, \boldsymbol{X}^I]$ to $[\boldsymbol{X}^P, \boldsymbol{X}_{t_i}^E]$, so that DiT only estimates the velocity of the edited region $\boldsymbol{v}_{t_i}^E$. However, since DiT contains attention layers that involve global token interactions, completely discarding the $\boldsymbol{X}^I$ and $\boldsymbol{X}_{t_i}^U$ inputs can gradually inject bias into the estimation of $\boldsymbol{v}_{t_i}^E$ during global attention. To compensate for this loss of information, we propose a Region-Instruction KV Cache (RIKVCache). Specifically, the input to DiT remains $[\boldsymbol{X}^P, \boldsymbol{X}_{t_i}^E]$, but within the attention layers of DiT, it is modified as follows:

$$softmax(\frac{[\boldsymbol{Q}_P, \boldsymbol{Q}_E] \cdot [\boldsymbol{K}_P, \boldsymbol{K}_E, \boldsymbol{K}_U^C, \boldsymbol{K}_I^C]^T}{\sqrt{d}}) \cdot [\boldsymbol{V}_P, \boldsymbol{V}_E, \boldsymbol{V}_U^C, \boldsymbol{V}_I^C]. \tag{6}$$

The lower corner labels $P$, $E$, $U$, and $I$ represent prompt token, edited region token, unedited region token, and instruction token, respectively. The superscript $C$ in the upper-right corner indicates that the value is taken from the cache of the previous complete computation. And the middle-lower part of Figure 3 visualizes this process. The feasibility of this approach is supported by the high similarity of the KV pairs between consecutive steps, as shown in Figure 2g and 2h.

**Adaptive Velocity Decay Cache.** As illustrated in the right part of Figure 1, although the trajectory of the edited region is curved, the velocities between consecutive timesteps are actually similar. Focusing on the intermediate denoising phase, we observe from Figure 2e that the velocity directions between adjacent steps are almost identical (cosine similarity approaches 1). At the same time, the magnitudes exhibit a gradual decay that varies across timesteps (Figure 2d). Based on this observation, we propose an adaptive velocity decay cache (AVDCache). Specifically, the AVDCache introduces a decay factor:

$$||\boldsymbol{v}_{t_i}||/||\boldsymbol{v}_{t_{i+1}}|| = (1 - \Delta t_{t_{i+1}, t_i}) \cdot \gamma_{t_i}. \tag{7}$$

Here, $(1 - \Delta t_{t_{i+1}, t_i})$ represents the sample-aware component under discrete Euler solver, while $\gamma_{t_i}$ represents the timestep-aware component. The solver entirely determines the former, while the latter is obtained by fitting on a randomly sampled dataset. Since the decay factor in Eq. 7 characterizes the intrinsic differences between diffusion model timesteps, we introduce the AVDCache criterion:

$$Criterion = 1 - \prod_{i=s}^{e} (1 - \Delta t_{t_{i+1}, t_i}) \cdot \gamma_{t_i}. \tag{8}$$

Here, $t_s$ and $t_e$ denote the start and end timesteps of the cache, respectively, while the criterion measures the cumulative error of this process. The decision of whether to apply the cache is made using a threshold $\delta$. The complete process is as follows:

$$\boldsymbol{v}_{t_i}^E = \begin{cases} DiT(\boldsymbol{X}_{t_i}^E, \boldsymbol{X}^P) & Criterion > \delta \\ \boldsymbol{v}_{t_s}^{E,C} \cdot \prod_{m=s}^{i} (1 - \Delta t_{t_{m+1}, t_m}) \cdot \gamma_{t_m} & else. \end{cases} \tag{9}$$

The right-lower part of Figure 3 visualizes this process. In fact, AVDCache is an improved version of the existing residual cache methods, with further details and analysis provided in the supplementary.

After the above process, we obtain the generated results for both the edited and unedited regions. We then re-gather these results according to their spatial positions to reconstruct the complete image tokens. It is worth noting that the similarity of the KV Cache decreases as the timestep increases. To address this issue, we periodically enforce full-image gathering at certain timesteps within the region-aware generation stage, performing a complete DiT computation to update the RIKVCache.

**Smooth Stage.** Small gaps may appear at the boundaries between edited and unedited regions after stitching. Although these gaps are often imperceptible in most cases, to ensure the generality of our method, we perform several steps of unaccelerated denoising on the merged full image to smooth these discontinuities. Empirically, two denoising steps are sufficient to eliminate the gaps effectively.

# 5 EXPERIMENT

## 5.1 EXPERIMENTAL SETTINGS

**Pretrained Model & Dataset.** We evaluate RegionE on three open-source state-of-the-art IIE models: Step1X-Edit-v1p1 (Liu et al., 2025b), FLUX.1 Kontext (Labs et al., 2025), and Qwen-Image-Edit (Wu et al., 2025). Step1X-Edit adopts a CFG (classifier-free guidance) (Ho & Salimans, 2022) scale of 6, FLUX.1 Kontext uses a scale of 2.5, and Qwen-Image-Edit applies a scale of 4. All models are evaluated with 28 sampling steps. For evaluation, we follow the dataset protocols described in the respective technical reports. Specifically, we use 606 image prompt pairs covering 11 tasks from GEdit-Bench English (Liu et al., 2025b) for Step1X-Edit and Qwen-Image-Edit, and 1026 image prompt pairs spanning five tasks from KontextBench (Labs et al., 2025) for FLUX.1 Kontext.

**Evaluation Metrics.** We design a comprehensive evaluation framework to assess both the quality and efficiency of IIE models. For quality assessment, we adopt two complementary approaches. First, we evaluate reconstruction quality by measuring deviations before and after acceleration, using PSNR (Zhao et al., 2024), SSIM (Wang & Bovik, 2002), and LPIPS (Zhang et al., 2018) as metrics. Second, we conduct an editing evaluation using vision–language models (VLMs), specifically GPT-4o, to assess image quality, semantic alignment, and overall performance (Ku et al., 2024), as shown in Table 1. Evaluation dimensions are denoted by the suffixes SC, PQ, and O, consistent with (Liu et al., 2025b) and (Wu et al., 2025). For efficiency evaluation, we report actual runtime latency as well as the relative speedup compared to the vanilla pretrained models.

**Baseline.** Currently, there are no acceleration methods designed explicitly for IIEmodels. Therefore, we adapt several effective acceleration techniques initially developed for diffusion models as baselines, since they are also applicable to diffusion-based IIE tasks. From the perspective of timestep redundancy, Steoskip performs larger jumps in the sampling steps, FORA (Selvaraju et al., 2024) employs block-level cache, and $\Delta$-DiT (Chen et al.) and TeaCache (Liu et al., 2025a) use residual cache. From the perspective of spatial redundancy, RAS (Liu et al., 2025c) and ToCa (Zou et al., 2024a) perform redundancy-reduction denoising at the token level.

**Implementation Details.** For all three models, RegionE uses six steps in the stabilization stage, enforces an update at step 16 in the region-aware generation stage, and adopts two steps in the smooth stage. For Step1X-Edit, FLUX.1 Kontext, and Qwen-Image-Edit, the segmentation thresholds $\eta$ of ARP are 0.88, 0.93, and 0.80, respectively, while the decision thresholds $\delta$ of AVDCache are 0.02, 0.04, and 0.03, respectively. Latency is measured on a single NVIDIA H800 GPU.

## 5.2 EXPERIMENTAL RESULTS ANALYSIS

We evaluate RegionE against several state-of-the-art acceleration methods on three prominent IIE models: Step1X-Edit, FLUX.1 Kontext, and Qwen-Image-Edit. Our evaluation encompasses quantitative metrics, efficiency measurements, and visualization, demonstrating that RegionE achieves a superior balance between acceleration and quality preservation. The quantitative results are shown in Table 1. Since both GEdit-Bench and KontextBench involve multiple editing tasks, the table reports results averaged over tasks, while the per-task quantitative results are provided in the supplementary.

**Deviation Analysis Compared to Pre-trained Models.** The Against Vanilla evaluation reveals RegionE's exceptional fidelity to original model outputs across all evaluation metrics, significantly outperforming competing acceleration methods. RegionE achieves the highest PSNR values: 30.520 (Step1X-Edit), 32.133 (FLUX.1 Kontext), and 31.115 (Qwen-Image-Edit), representing substantial improvements of 2-4 over the next-best methods, indicating minimal pixel-level deviation from the original outputs. The SSIM scores of 0.939, 0.917, and 0.937 demonstrate superior preservation of structural coherence across different model architectures. In contrast, the LPIPS scores of 0.054, 0.057, and 0.046 represent 25-50% improvements over competing methods. This consistent performance across three diverse model architectures validates RegionE's architectural agnosticism. RegionE consistently maintains stable, high-quality results.

**GPT-4o Editing Quality Assessment & User Study**. The GPT-4o evaluation provides additional quality validation through automated semantic and perceptual analysis across three dimensions, consistently demonstrating RegionE's superior performance. For semantic consistency (G-SC), RegionE achieves scores of 7.552, 7.278, and 8.242, matching or exceeding original models while maintaining

Table 1: Comparison of editing quality and efficiency between RegionE and the baseline. All the evaluations are carried out on a single NVIDIA H800 GPU. (S) denotes the strategy for reducing spatial redundancy, while (T) denotes the strategy for reducing temporal redundancy.

| Model | Type | Against Vanilla | | | GPT-4o Score | | | Efficiency | |
|---|---|---|---|---|---|---|---|---|---|
| | | PSNR↑ | SSIM↑ | LPIPS↓ | G-SC↑ | G-PQ↑ | G-O↑ | Latency (s)↓ | Speedup↑ |
| **Step1X-Edit** (Liu et al., 2025b) | | - | - | - | 7.479 | 7.466 | 6.906 | 27.945 | 1.000 |
| + Stepskip | T | 26.719 | 0.898 | 0.096 | 7.491 | 7.343 | 6.880 | 12.299 | 2.272 |
| + FORA (Selvaraju et al., 2024) | T | 22.126 | 0.835 | 0.178 | 6.078 | **7.588** | 5.863 | 14.330 | 1.950 |
| + Δ-DiT (Chen et al.) | T | 24.659 | 0.874 | 0.122 | 7.432 | 7.233 | 6.795 | 12.728 | 2.196 |
| + TeaCache (Liu et al., 2025a) | T | 28.262 | 0.924 | 0.072 | 7.455 | 7.361 | 6.866 | 11.212 | 2.493 |
| + RAS (Liu et al., 2025c) | S | 26.819 | 0.892 | 0.100 | 7.339 | 7.072 | 6.615 | 15.239 | 1.834 |
| + ToCa (Zou et al., 2024a) | S | 24.699 | 0.844 | 0.152 | 7.185 | 6.705 | 6.350 | 22.149 | 1.262 |
| + Ours (RegionE) | T & S | **30.520** | **0.939** | **0.054** | **7.552** | 7.405 | **6.948** | **10.865** | **2.572** |
| **FLUX.1 Kontext** (Labs et al., 2025) | | - | - | - | 7.197 | **6.963** | 6.497 | 14.682 | 1.000 |
| + Stepskip | T | 26.199 | 0.838 | 0.123 | 7.126 | 6.938 | 6.463 | 8.512 | 1.725 |
| + FORA (Selvaraju et al., 2024) | T | 24.685 | 0.809 | 0.146 | 7.085 | 6.897 | 6.383 | 7.497 | 1.958 |
| + Δ-DiT (Chen et al.) | T | 20.227 | 0.723 | 0.225 | 7.055 | 6.918 | 6.411 | 6.751 | 2.175 |
| + TeaCache (Liu et al., 2025a) | T | 28.307 | 0.869 | 0.097 | 7.233 | 6.846 | 6.455 | 6.203 | 2.367 |
| + RAS (Liu et al., 2025c) | S | 26.217 | 0.829 | 0.132 | 7.216 | 6.785 | 6.460 | 8.219 | 1.786 |
| + ToCa (Zou et al., 2024a) | S | 23.906 | 0.767 | 0.192 | 6.985 | 6.589 | 6.237 | 11.299 | 1.299 |
| + Ours (RegionE) | T & S | **32.133** | **0.917** | **0.057** | **7.278** | 6.953 | **6.538** | **6.096** | **2.409** |
| **Qwen-Image-Edit** (Wu et al., 2025) | | - | - | - | 8.242 | 7.948 | 7.700 | 32.125 | 1.000 |
| + Stepskip | T | 28.439 | 0.892 | 0.077 | 8.090 | 7.875 | 7.572 | 17.555 | 1.830 |
| + FORA (Selvaraju et al., 2024) | T | 26.508 | 0.863 | 0.098 | 8.032 | 7.760 | 7.501 | 17.815 | 1.803 |
| + Δ-DiT (Chen et al.) | T | 25.020 | 0.821 | 0.116 | 7.964 | 7.718 | 7.417 | 17.470 | 1.839 |
| + TeaCache (Liu et al., 2025a) | T | 28.314 | 0.900 | 0.075 | 8.084 | 7.841 | 7.563 | 16.445 | 1.954 |
| + RAS (Liu et al., 2025c) | S | 27.251 | 0.879 | 0.090 | 8.152 | 7.680 | 7.515 | 22.327 | 1.439 |
| + ToCa (Zou et al., 2024a) | S | OOM | OOM | OOM | OOM | OOM | OOM | OOM | OOM |
| + Ours (RegionE) | T & S | **31.115** | **0.937** | **0.046** | **8.242** | **7.968** | **7.731** | **15.604** | **2.059** |

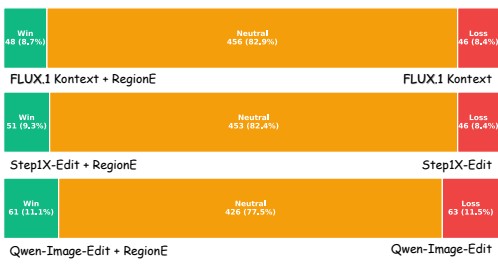

Figure 4: User study results for the RegionE.

substantial acceleration, with Qwen-Image-Edit showing perfect preservation (8.242) despite 2.059× speedup. The perceptual quality (G-PQ) scores of 7.405, 6.953, and 7.968 consistently outperform competing acceleration methods by 0.1 to 0.3 points, demonstrating the practical preservation of visual coherence through region-aware processing. Overall quality (G-O) scores of 6.948, 6.538, and 7.731 provide holistic assessment validation, with the alignment between GPT-4o assessments and quantitative metrics (PSNR, SSIM, LPIPS) strengthening confidence in RegionE's comprehensive quality preservation across multiple evaluation dimensions and providing additional evidence of the hybrid temporal-spatial optimization approach's effectiveness. We also conducted a user study, and the results are shown in Figure 4. The findings indicate that participants had difficulty discerning whether the edited images were accelerated using RegionE, further validating the high-fidelity capabilities of RegionE.

**Efficiency Analysis.** RegionE demonstrates substantial efficiency gains while maintaining superior quality, achieving an optimal balance between acceleration and performance preservation with impressive results across all evaluated models. The method achieves speedups of 2.572×, 2.409×, and 2.059× across Step1X-Edit, FLUX.1 Kontext, and Qwen-Image-Edit respectively, translating to significant absolute latency reductions: from 27.945s to 10.865s, from 14.682s to 6.096s, and from 32.125s to 15.604s respectively. RegionE occupies the optimal position on the efficiency-quality curve, maintaining the highest quality metrics while achieving competitive or superior acceleration compared to methods that sacrifice substantial quality for higher speedups.

**Visualization.** Figure 5 presents partial visualizations of different acceleration methods on Step1X-Edit. Among the baselines, RegionE produces edited outputs closest to the vanilla setting at higher speedups, preserving both details and contours. The last column shows ARP predictions of spatial regions in RegionE, where unedited regions are masked. These masked regions closely match human perception. Additional visualizations for other tasks and models are provided in the supplementary.

Table 2: Ablation study on cache design and stage design in RegionE.

| Variant | | Against Vanilla | | | GPT-4o Score | | | Efficiency | |
| --- | --- | --- | --- | --- | --- | --- | --- | --- | --- |
| | | PSNR↑ | SSIM↑ | LPIPS↓ | G-SC↑ | G-PQ↑ | G-O↑ | Latency (s)↓ | Speedup↑ |
| **RegionE** | | 30.520 | 0.939 | 0.054 | 7.552 | 7.405 | 6.948 | 10.865 | 2.572 |
| **Cache Design** | w/o RIKVCache | 22.868 | 0.822 | 0.207 | 5.997 | 5.389 | 5.191 | 10.223 | 2.734 |
| | w/o AVDCache | 31.139 | 0.946 | 0.046 | 7.570 | 7.482 | 7.023 | 16.122 | 1.733 |
| **Stage Design** | w/o STS | 21.441 | 0.814 | 0.161 | 7.045 | 6.758 | 6.325 | 7.149 | 3.909 |
| | w/o SMS | 28.857 | 0.904 | 0.085 | 7.456 | 7.207 | 6.773 | 9.766 | 2.862 |
| | w/o Forced Step | 28.452 | 0.915 | 0.080 | 7.536 | 7.305 | 6.925 | 10.204 | 2.739 |

Figure 5: Examples of edited images by RegionE and baseline on Step1X-Edit-v1p1.

## 5.3 ABLATION STUDY

We conduct ablation studies to investigate the contributions of different components in RegionE, primarily on the Step1X-Edit-v1p1. The quantitative results are summarized in Table 2.

**Cache Design.** We propose two key components: RIKVCache and AVDCache. Removing RIKV-Cache, i.e., performing local attention within the edited region without injecting instruction information or context from the unedited region, results in a 2.734× speed-up. However, this comes at a significant cost to editing quality, with PSNR dropping from 30.520 to 22.868 and G-O decreasing from 6.948 to 5.191. This demonstrates that global context supervision is crucial even during region generation. In contrast, removing AVDCache results in a slight improvement in editing quality (G-O increases from 6.948 to 7.023), but without eliminating redundancy across timesteps, the acceleration is limited to 1.733. This indicates that AVDCache significantly improves inference efficiency with minimal degradation in quality.

**Stage Design.** We introduce two auxiliary stages: Stabilization Stage (STS) and Smooth Stage (SMS), as well as a forced step in the region-aware generation stage (RAGS). Removing STS causes substantial drops in editing quality (PSNR: 30.520 → 21.441, LPIPS: 0.054 → 0.161, G-O: 6.948 → 6.325). As discussed in Section 4, STS addresses the instability in speed estimation, and skipping it results in degraded performance. Removing SMS leads to smaller declines in both pixel-level (PSNR: 30.520 → 28.857, SSIM: 0.939 → 0.904) and perceptual metrics (G-O: 6.948 → 6.773), reflecting its role in bridging the gap between edited and unedited regions. Finally, when the forced step in RAGS was removed, since its role was to mitigate the decay of KV similarity over time, its removal led to a 2-point drop in PSNR, further validating its necessity.

**Sensitivity of Parameters $\delta$ and $\eta$.** The parameter $\delta$ controls the proportion of cached timesteps. The parameter $\eta$ is used to distinguish between edited and unedited regions. We evaluate 25 different

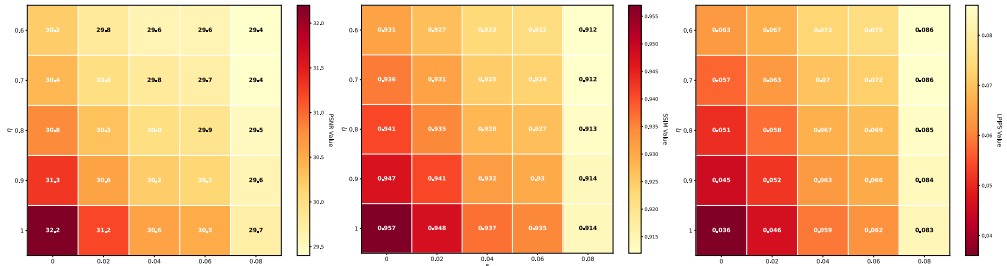

Figure 6: Sensitivity analysis of hyperparameters $\delta$ and $\eta$ performed on Step1X-Edit-v1p1.

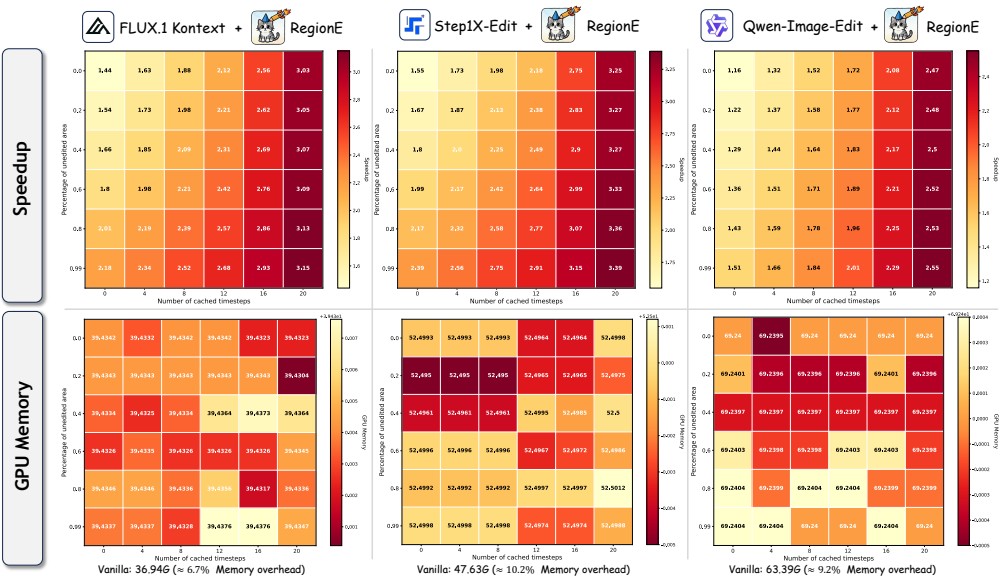

Figure 7: Speedup and GPU memory usage at different levels of spatiotemporal redundancy.

combinations of $\delta$ and $\eta$ on the Step1X-Edit, and the quantitative results of editing quality are shown in Figure 6. The results indicate that: (1) as $\delta$ increases, more timesteps are skipped and editing quality deteriorates; and (2) as $\eta$ increases, a larger portion of the image is considered edited, resulting in slower generation due to the increased area requiring local synthesis, but improved editing quality.

**Speedup and GPU Memory Across Redundancy Levels.** Figure 7 summarizes the speedup and memory consumption of RegionE under varying redundancy levels. The horizontal axis represents the number of cached timesteps, and the vertical axis denotes the proportion of unedited regions, evaluated on 1024×1024 images. As the edited region shrinks and more timesteps are skipped, RegionE yields higher speedups, reaching up to 3.15×, 3.39×, and 2.55× on FLUX.1 Kontext, Step1X-Edit, and Qwen-Image, respectively. Memory usage remains largely unaffected across redundancy levels, with RegionE incurring only 6%–10% additional overhead compared to the vanilla setting.

## 6 CONCLUSION

Inspired by temporal and spatial redundancy in IIE, we propose RegionE, an adaptive, region-aware generation framework that accelerates the IIE process. Specifically, we perform early prediction on spatial regions using ARP and combine it with RIKVCache for region-wise editing to reduce spatial redundancy. We also use AVDCache to minimize temporal redundancy. Experiments show that RegionE achieves 2.57×, 2.41×, and 2.06× end-to-end speedups on Step1X-Edit and FLUX.1 Kontext, and Qwen-Image-Edit, respectively, while maintaining minimal bias (PSNR 30.52–32.13) and negligible quality loss (GPT-4o evaluation results remain comparable). These results demonstrate the effectiveness of RegionE in reducing redundancy in IIE.

ACKNOWLEDGMENTS

This work is supported by Shanghai Natural Science Foundation (No. 23ZR1402900), Shanghai Science and Technology Commission Explorer Program Project (24TS1401300).

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

## RegionE: Adaptive Region-Aware Generation for Efficient Image Editing

### Supplementary Material

We organize the supplementary material as follows:

- Section A: Pseudocode of RegionE
- Section B: Analysis of Adaptive Velocity Decay Cache
- Section C: One-Step Prediction from Different Denoising Timesteps
- Section D: Discussion on Using AVDCache During the Stabilization Stage
- Section E: Discussion on High-Resolution Image Editing
- Section F: Discussion on Editing Boundaries
- Section G: Discussion on Multi-Region Editing
- Section H: Discussion on Global Editing
- Section I: Discussion on Bad Cases
- Section J: Experimental Setup of the User Study
- Section K: Per-Task Visualization Results in the Benchmark
- Section L: Per-Task Quantitative Results in the Benchmark
- Section M: Statement on LLM Usage

# A    PSEUDOCODE OF REGIONE

---

**Algorithm 1** RegionE: Adaptive Region-Aware Generation for Efficient Image Editing

---

**Input:** Diffusion transformer $\Phi(\cdot)$, sampling step $T$, insturction image $\boldsymbol{X}^I$, text tokens $\boldsymbol{X}^P$, random noise $\boldsymbol{X}_T$, total steps in stabilization stage $t^{st}$, total steps in smooth stage $t^{sm}$, threshold of adaptive region partition $\eta$, threshold of adaptive velocity decay cache $\delta$, sorted forced steps list $t_f\_list$.

1: // **Initialization**
2: RIKVCache $\mathcal{C}_{\mathcal{KV}}$ = None, RIKVCache flag $f = (False, False)$; AVDCache $\mathcal{C}_{\mathcal{A}}$=None;
3: Accumulative Error $e = 0$; $t_f\_list.insert(0, T - t^{st})$; $t_f\_list.insert(-1, t^{sm} - 1)$;
4: // **Stabilization Stage**
5: **for** $i \leftarrow T$ to $T - t^{st}$ **do**
6:     **if** $i == T - t^{st}$ **then**
7:         $f[0] = True \triangleright$ *first dimension represents storing, second dimension represents retrieving*
8:     **end if**
9:     $\boldsymbol{v}_{t_i}, \mathcal{C}_{\mathcal{KV}} = \Phi([\boldsymbol{X}^P, \boldsymbol{X}_{t_i}, \boldsymbol{X}^I], \mathcal{C}_{\mathcal{KV}}, f)$
10:     $\boldsymbol{X}_{t_{i-1}} = \boldsymbol{X}_{t_i} - (t_i - t_{i-1}) \cdot \boldsymbol{v}_{t_i}$
11: **end for**
12: // **Region-Aware Generation Stage**
13: $\triangleright$ *Adaptive Region Partition*
14: $\hat{\boldsymbol{X}}_0 = \boldsymbol{X}_{t_{T-t^{st}}} - \boldsymbol{v}_{T-t^{st}+1} \cdot t_{T-t^{st}}$
15: $E_{index}, U_{index} = Erosion\_\&\_Dilate(cos(\hat{\boldsymbol{X}}_0, \boldsymbol{X}^I) > \eta)$
16: $\triangleright$ *Region-Aware Generation*
17: **for** $i \leftarrow 0$ to $len(t_f\_list) - 2$ **do**
18:     $prev = t_f\_list[i]; next = t_f\_list[i+1]$
19:     $\boldsymbol{X}_{t_{prev}}^E = \boldsymbol{X}_{t_{prev}}[E_{index}]; \boldsymbol{X}_{t_{prev}}^U = \boldsymbol{X}_{t_{prev}}[U_{index}]$
20:     $\hat{\boldsymbol{X}}_{t_{next+1}}^U = \boldsymbol{X}_{t_{prev}}^U - \boldsymbol{v}_{t_{prev}+1}^U \cdot (t_{prev} - t_{next+1}) \triangleright$ *one-step estimation for unedited region*
21:     $f[0] = False, f[1] = True \triangleright$ *iteritive denoising for edited region*
22:     **for** $j \leftarrow prev$ to $next + 1$ **do**
23:         $\triangleright$ *Adaptive Velocity Decay Cache*
24:         Calculate $e$ according to Eq.8
25:         **if** $e > \delta$ **then**
26:             $\boldsymbol{v}_{t_j}^E, \mathcal{C}_{\mathcal{KV}} = \Phi([\boldsymbol{X}^P, \boldsymbol{X}_{t_j}^E, \boldsymbol{X}^I], \mathcal{C}_{\mathcal{KV}}, f)$
27:             $\mathcal{C}_{\mathcal{A}} = \boldsymbol{v}_{t_j}^E$
28:             $\boldsymbol{X}_{t_{j-1}}^E = \boldsymbol{X}_{t_j}^E - (t_j - t_{j-1}) \cdot \boldsymbol{v}_{t_j}^E$
29:         **else**
30:             $\boldsymbol{v}_{t_j}^E = \mathcal{C}_{\mathcal{A}}*$decay factor according to Eq.7
31:         **end if**
32:     **end for**
33:     $\boldsymbol{X}_{t_{next+1}} = gather(\boldsymbol{X}_{t_{next}}^U, \boldsymbol{X}_{t_{next+1}}^E)$
34:     $f[0] = True, f[1] = False$
35:     $\boldsymbol{v}_{t_{next+1}}, \mathcal{C}_{\mathcal{KV}} = \Phi([\boldsymbol{X}^P, \boldsymbol{X}_{t_{next+1}}, \boldsymbol{X}^I], \mathcal{C}_{\mathcal{KV}}, f)$
36:     $\boldsymbol{X}_{t_{next}} = \boldsymbol{X}_{t_{next+1}} - (t_{next} - t_{next+1}) \cdot \boldsymbol{v}_{t_{next+1}}$
37: **end for**
38: // **Smooth Stage**
39: $f[0] = False, f[1] = False$
40: **for** $i \leftarrow t^{sm} - 1$ to $1$ **do**
41:     $\boldsymbol{v}_{t_i}, \mathcal{C}_{\mathcal{KV}} = \Phi(\boldsymbol{X}_{t_i}, \mathcal{C}_{\mathcal{KV}}, f)$
42:     $\boldsymbol{X}_{t_{i-1}} = \boldsymbol{X}_{t_i} - (t_i - t_{i-1}) \cdot \boldsymbol{v}_{t_i}$
43: **end for**
**Output:** Target image after editing $\boldsymbol{X}_0$

---

# B    ANALYSIS OF ADAPTIVE VELOCITY DECAY CACHE

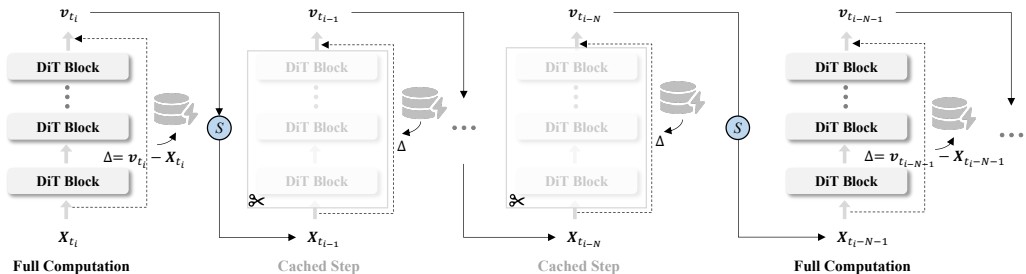

Figure 8: Pipeline Based on Residual Cache.

In current research on diffusion model caching, many studies focus on residual-based caches (Chen et al.; Liu et al., 2025a; Zhou et al., 2025; Bu et al., 2025), which store the $\Delta$ shown in Figure 8. Based on the sampling formula in Equation 4 and the definition of caching, we can derive the following expression:

$$
\begin{cases}
\boldsymbol{X}_{t_{i-1}} = \boldsymbol{X}_{t_i} - (t_i - t_{i-1}) \cdot \boldsymbol{v}_{t_i} \\
\quad\;\; \Delta = \boldsymbol{v}_{t_i} - \boldsymbol{X}_{t_i} \\
\quad \boldsymbol{v}_{t_{i-1}} = \boldsymbol{X}_{t_{i-1}} + \Delta
\end{cases} .
\tag{10}
$$

It can be solved as:

$$
\boldsymbol{v}_{t_{i-1}} = [1 - (t_i - t_{i-1})] \cdot \boldsymbol{v}_{t_i}.
\tag{11}
$$

Similarly, for the timestep $t_{i-2}$, we have:

$$
\boldsymbol{v}_{t_{i-2}} = [1 - (t_{i-1} - t_{i-2})] \cdot \boldsymbol{v}_{t_{i-1}}.
\tag{12}
$$

Therefore, if we perform $N$ steps of residual caching, as illustrated in Figure 8, we can obtain:

$$
\begin{aligned}
\boldsymbol{v}_{t_{i-N}} &= \prod_{m=1}^{N} [1 - (t_{i-m+1} - t_{i-m})] \cdot \boldsymbol{v}_{t_i} \\
&= \underbrace{\prod_{m=1}^{N} [1 - \Delta t_{i-m+1,i-m}]}_{\text{Determined by Solver}} \cdot \boldsymbol{v}_{t_i}.
\end{aligned}
\tag{13}
$$

This further indicates that the current residual cache and the velocity cache are equivalent. Since $\Delta t_{i-m+1,i-m}$ is a minimal value approaching zero, the coefficient before $\boldsymbol{v}_{t_i}$ is less than one. Therefore, it can be seen that the current residual cache is essentially a decayed form of the velocity cache. Furthermore, we observe that the solver determines the decay coefficient in Equation 13. However, as shown in Figure 2d, the decay of velocity exhibits a timestep-dependent behavior. To account for this, we introduce an external timestep correction coefficient $\gamma_{t_i}$. Notably, the AVDCache proposed in this paper reduces to Equation 13 when the correction coefficient $\gamma_{t_i}$ equals 1.

# C    ONE-STEP PREDICTION FROM DIFFERENT DENOISING TIMESTEPS

In the manuscript, we demonstrate that the generative trajectories differ significantly between edited and unedited regions (see Figures 1 and 2). Specifically, trajectories in unedited regions are nearly linear, allowing early-stage velocity to provide reliable one-step estimates of the multi-step denoised images, including the final output. In contrast, edited regions exhibit noticeably curved trajectories, making the final result substantially harder to predict from early timesteps. To further provide intuitive and comprehensive evidence for this observation, we visualize full-image one-step predictions of the final edited image using velocities from different denoising timesteps (Figures 9 and 10). The results clearly show that unedited regions converge relatively early, whereas edited regions do

not. Consistent with our trajectory analysis, the near-linear dynamics in unedited regions enable accurate early prediction, whereas the complex, curved trajectories in edited regions require iterative refinement. This empirical evidence further supports the need to treat the two regions differently during denoising.

**Prompt：Add a hat to the cat.**

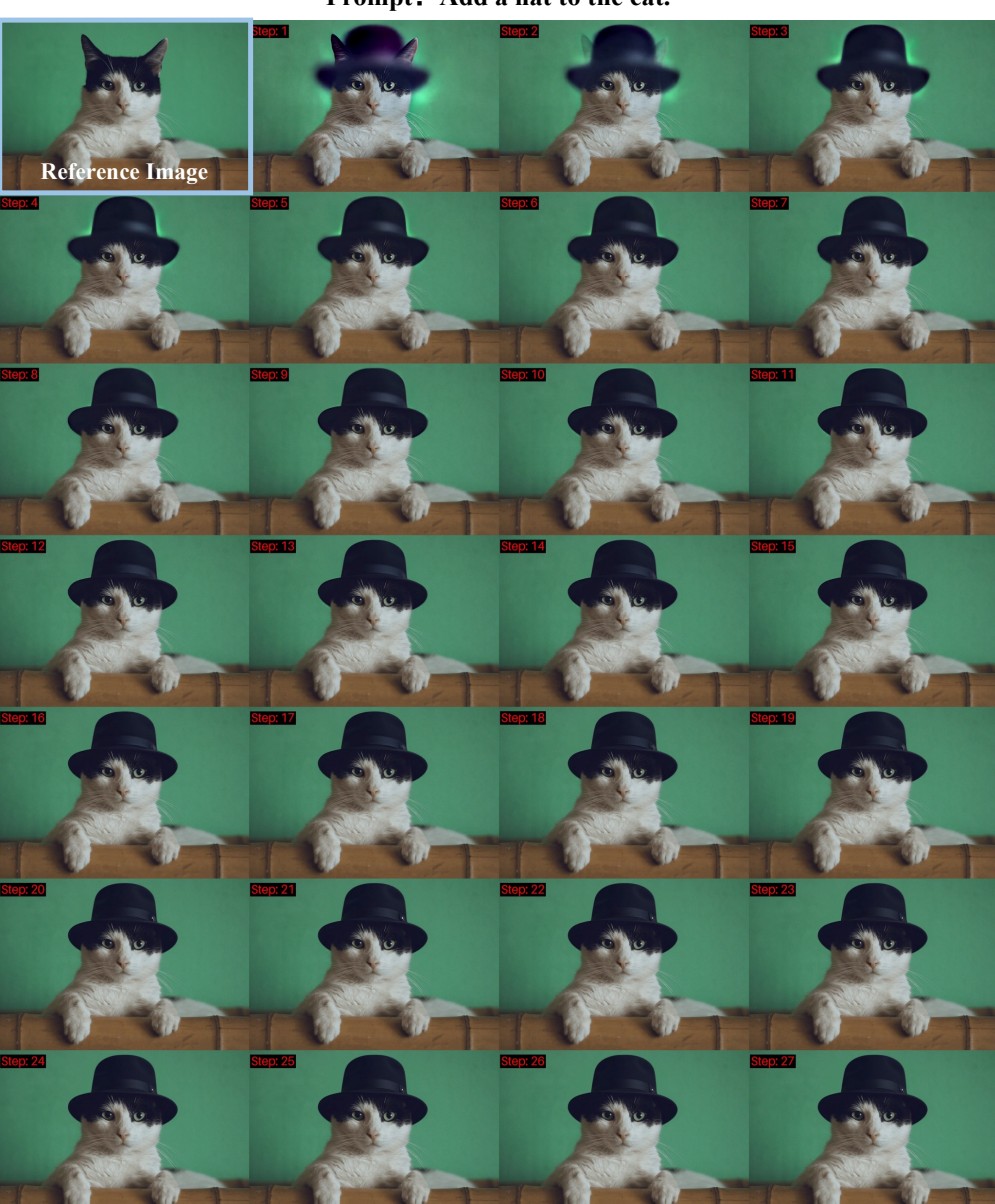

Figure 9: One-step predictions of the final edited image using velocity from different denoising timesteps.

## D DISCUSSION ON USING AVDCACHE DURING THE STABILIZATION STAGE

In RegionE, AVDCache is used exclusively in the Region-Aware Generation Stage. This design choice is motivated by the following considerations:

The first stage of RegionE is the Stabilization Stage. We do not apply caching in this stage for four reasons. (a) As discussed in Section 4 of the manuscript, the input in this stage has a low

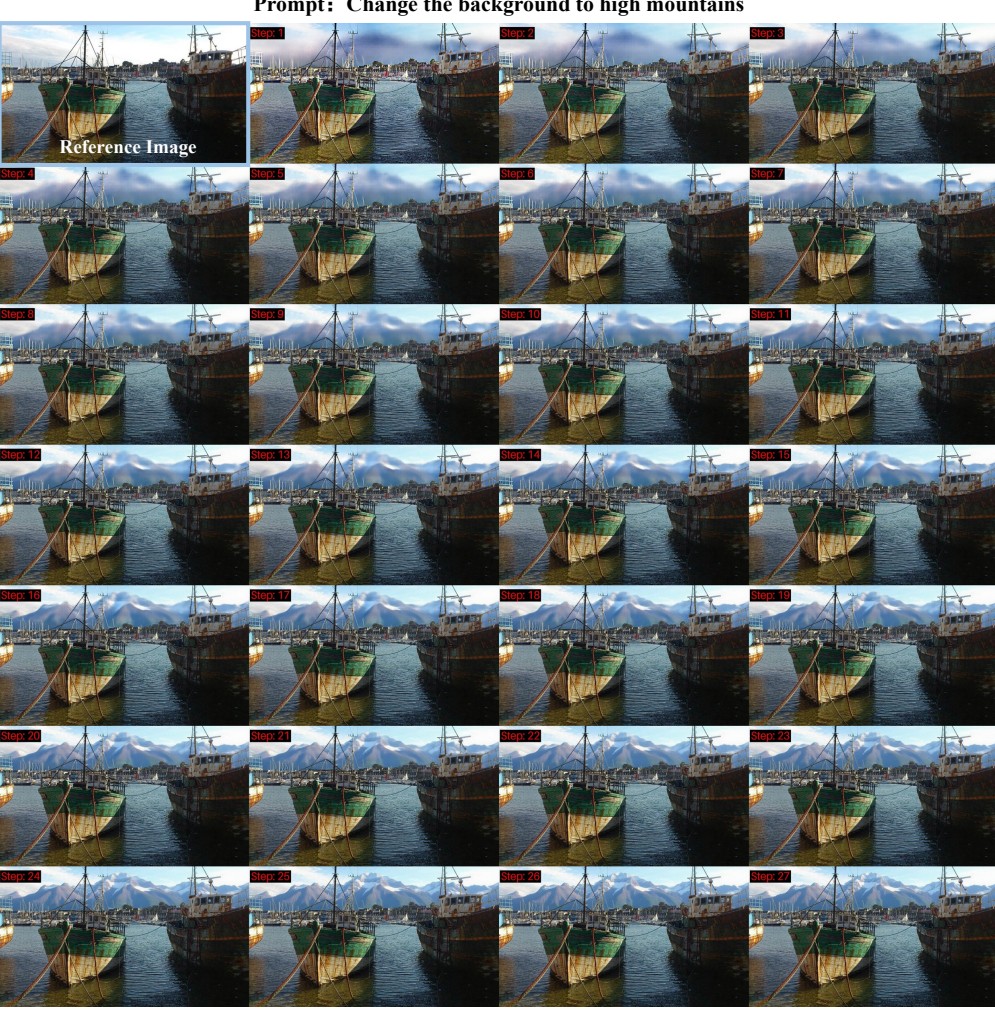

Figure 10: One-step predictions of the final edited image using velocity from different denoising timesteps.

signal-to-noise ratio, and the DiT predictions are inherently unstable, making it unsuitable for acceleration techniques such as caching. (b) Velocity similarity between consecutive steps is very low at the beginning, and since this stage is responsible for shaping the coarse structure of the image, introducing caching would harm generation quality. (c) This stage concludes with the separation of edited and unedited regions; thus, avoiding any loss during this stage is crucial. (d) Prior studies have also emphasized avoiding efficient methods at early timesteps, such as SVG (Xi et al., 2025), ViDiT-Q (Zhao et al., 2025b), and others.

For completeness, we also applied AVDCache to the Stabilization Stage, and the quantitative results are shown in Table 3. We observe that applying AVDCache in the STS yields higher speedups but also leads to a noticeable degradation in editing quality. Specifically, PSNR largely drops by 1.91, SSIM decreases by 0.013, LPIPS worsens by 0.16, and G-O declines by 0.066. To achieve a better balance between generation quality and efficiency, we therefore choose not to apply AVDCache in the STS.

## E    DISCUSSION ON HIGH-RESOLUTION IMAGE EDITING

In the main experiments, we focus on 1k-resolution reference images since Step1X-Edit, FLUX.1 Kontext, and Qwen-Image-Edit are all native 1k-resolution editing models. Here, we provide a

Table 3: Quantitative impact of applying AVDCache in the Stabilization Stage (STS).

| Model | Against Vanilla | | | GPT-4o Score | | | Efficiency | |
|---|---|---|---|---|---|---|---|---|
| | PSNR↑ | SSIM↑ | LPIPS↓ | G-SC↑ | G-PQ↑ | G-O↑ | Latency (s)↓ | Speedup↑ |
| **Step1X-Edit** | - | - | - | 7.479 | 7.466 | 6.906 | 27.945 | 1.000 |
| **+ Ours wo STS Cache** | 30.520 | 0.939 | 0.054 | 7.552 | 7.405 | 6.948 | 10.865 | 2.572 |
| **+ Ours w STS Cache** | 28.610 | 0.926 | 0.070 | 7.455 | 7.395 | 6.882 | 8.583 | 3.256 |

preliminary evaluation of RegionE on high-resolution image editing, as shown in Figure 11. High-resolution images contain more tokens after tokenization, resulting in greater spatial redundancy, which allows RegionE to achieve higher acceleration. The results in Figure 11 demonstrate both high fidelity and increased speedup, further validating this observation. Since there is currently no suitable benchmark for high-resolution image editing, we do not report quantitative results on a dataset.

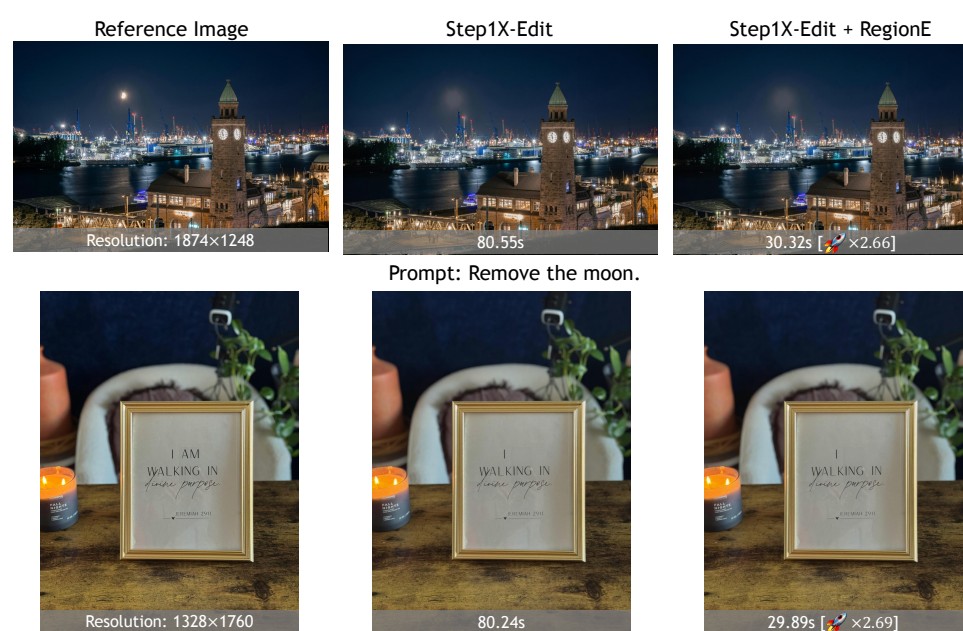

Figure 11: Visualization results in high-resolution image editing scenarios.

## F  DISCUSSION ON EDITING BOUNDARIES

Image editing with RegionE does not introduce boundary artifacts. This can be attributed to the following reasons: a) During local generation in the edited region, RegionE uses RIKVCache, which allows the Attention computation to access global key-value information. As a result, the edited region maintains awareness of the entire image, and only extremely minor boundary artifacts may occur at this stage. b) The final stage in RegionE is the Smooth Stage, which effectively eliminates any subtle boundary artifacts between edited and unedited regions. Two randomly selected visualization examples are shown in Figure 12, where the boundaries between edited and unedited regions are imperceptible.

## G  DISCUSSION ON MULTI-REGION EDITING

In Figure 5, we show generation results when the edited region is contiguous. In practice, whether the edited region is contiguous or dispersed does not affect RegionE's performance. This is because RegionE leverages RIKVCache, where the local query (Q) accesses global key-value (KV) information during Attention computation. Consequently, even dispersed edited regions attend to the same

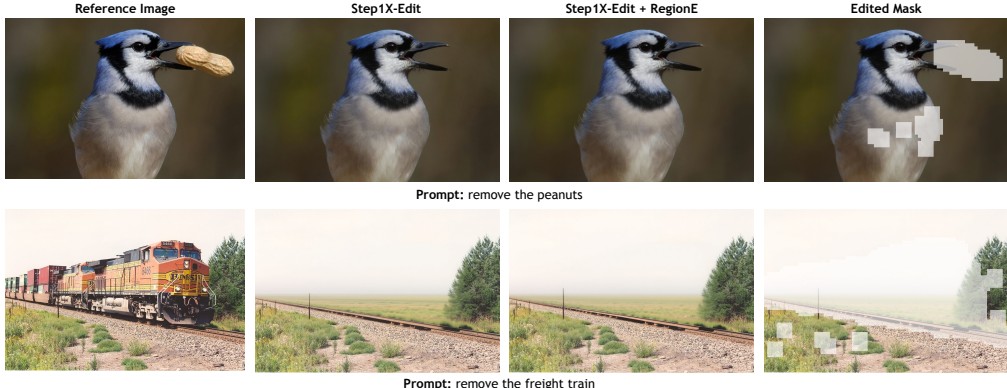

Figure 12: Visualization of image editing boundaries.

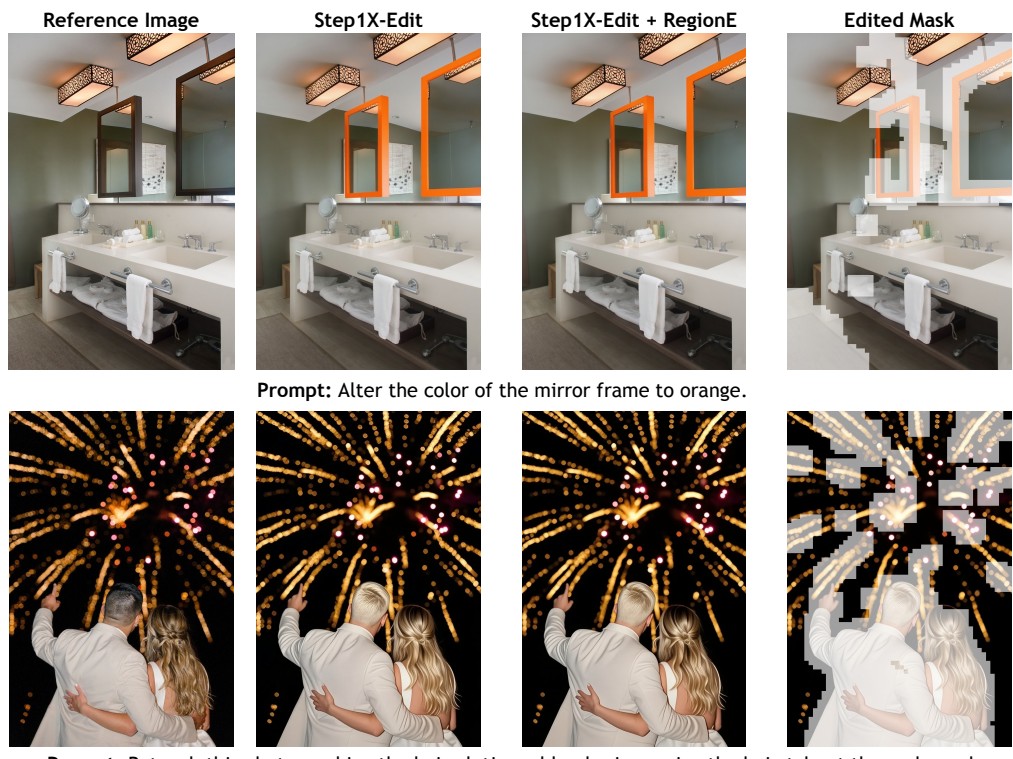

Figure 13: Visualization in multi-region image editing scenarios.

global context, avoiding significant computational bias. Figure 13 visualizes several examples with dispersed edited regions, demonstrating that RegionE achieves accelerated editing while maintaining high fidelity.

## H   DISCUSSION ON GLOBAL EDITING

In practical applications, fully global editing scenarios also occur. In such cases, the spatial redundancy in the editing task is low, and RegionE primarily exploits redundancy across timesteps to accelerate the editing process. Figure 14 shows an example of this type of task, demonstrating that RegionE can still achieve high-fidelity generation.

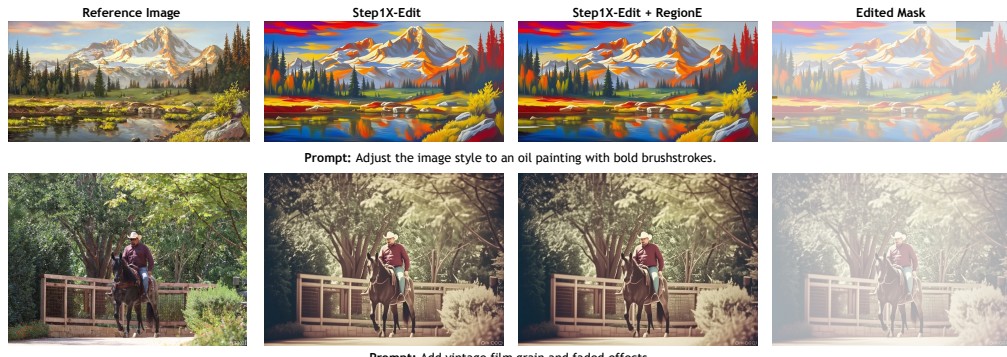

Figure 14: Visualization results in global image editing scenarios.

# I   DISCUSSION ON BAD CASES

At higher speedup, RegionE may produce some rare bad cases. Upon reviewing the entire dataset, we found that these few instances typically involve minor generation deviations that do not affect instruction adherence. As shown in Figure 15, in the first example, the color of the top corner slightly deviates, and in the second example, the shape of the ceramic shows a small discrepancy. However, these deviations do not compromise the overall adherence to the editing instructions.

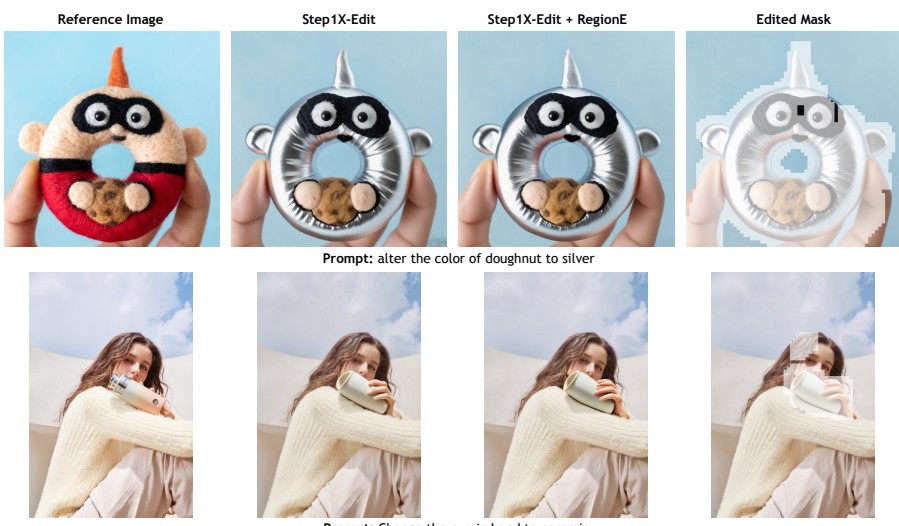

Figure 15: Visualization of failure cases.

# J   EXPERIMENTAL SETUP OF THE USER STUDY

In this section, we provide a detailed description of the user study setup. For evaluating RegionE on Step1X-Edit and Qwen-Image-Edit, we selected a total of 11 tasks from GEdit-Bench, randomly sampling 5 image–instruction pairs per task, resulting in 55 samples. For FLUX.1 Kontext, we selected 5 tasks from Kontext Bench, randomly sampling 11 image–instruction pairs per task, also totaling 55 samples.

After constructing the evaluation sets, we generated edited images using the base models both with and without RegionE, and saved the corresponding outputs. We then collected votes from 10 participants, who were asked to choose the image with higher quality and better instruction adherence. The order of the images was randomized, and participants were unaware of which method was used

for each image. If the two images were similar, participants could select a neutral option. Finally, the scores for the two methods were aggregated. The layout of the questionnaire is shown in Figure 16.

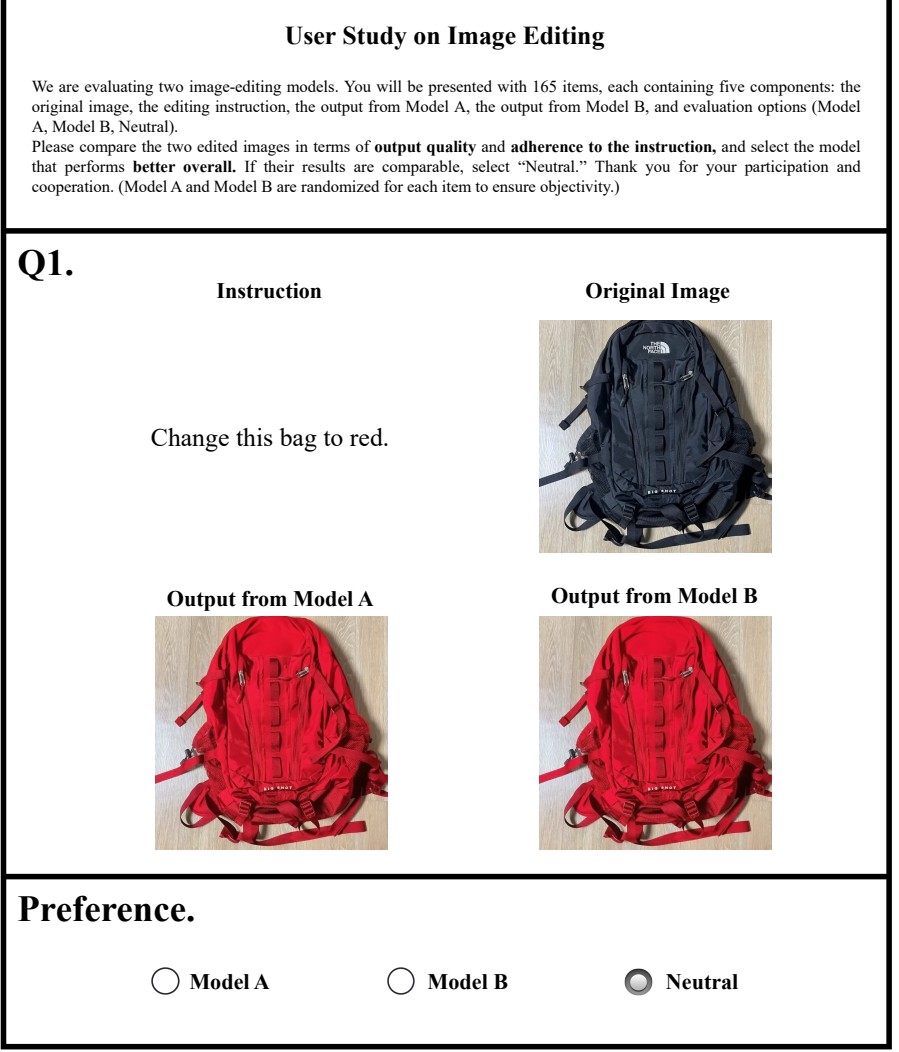

Figure 16: Thumbnails from the user study questionnaire.

## K  PER-TASK VISUALIZATION RESULTS IN THE BENCHMARK

Due to space limitations, we put the visualization results of some tasks in the manuscript. Here, we provide a visual comparison of additional tasks and models. Figure 17 and Figure 18 show the visualization results of 11 tasks on Step1X-Edit. Figure 20 and Figure 21 show the visualization results of 11 tasks on Qwen-Image-Edit. Figure 19 show the visualization results of 5 tasks on FLUX.1 Kontext.

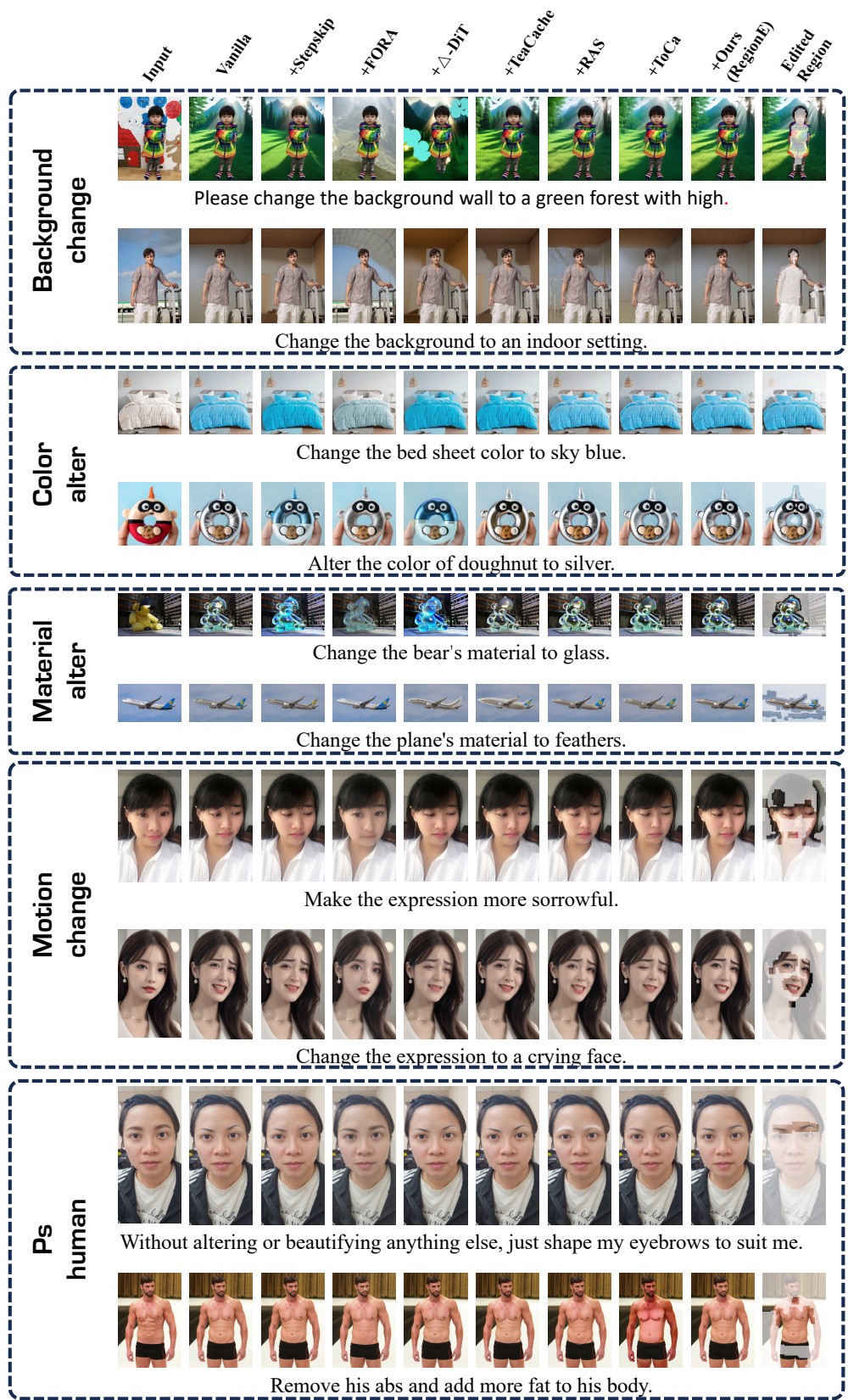

Figure 17: Examples of edited images by RegionE and baseline on Step1X-Edit-v1p1.

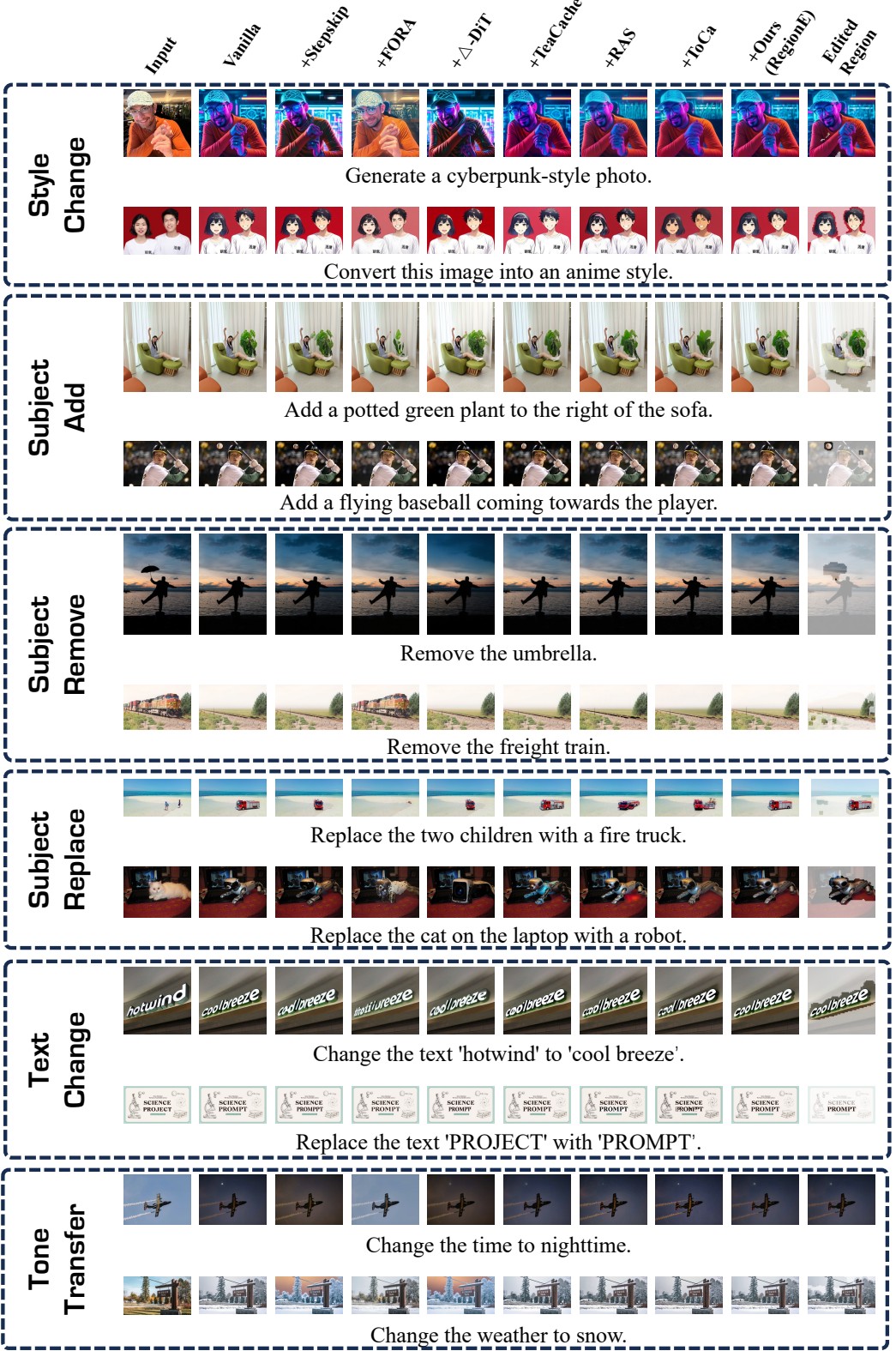

Figure 18: Examples of edited images by RegionE and baseline on Step1X-Edit-v1p1.

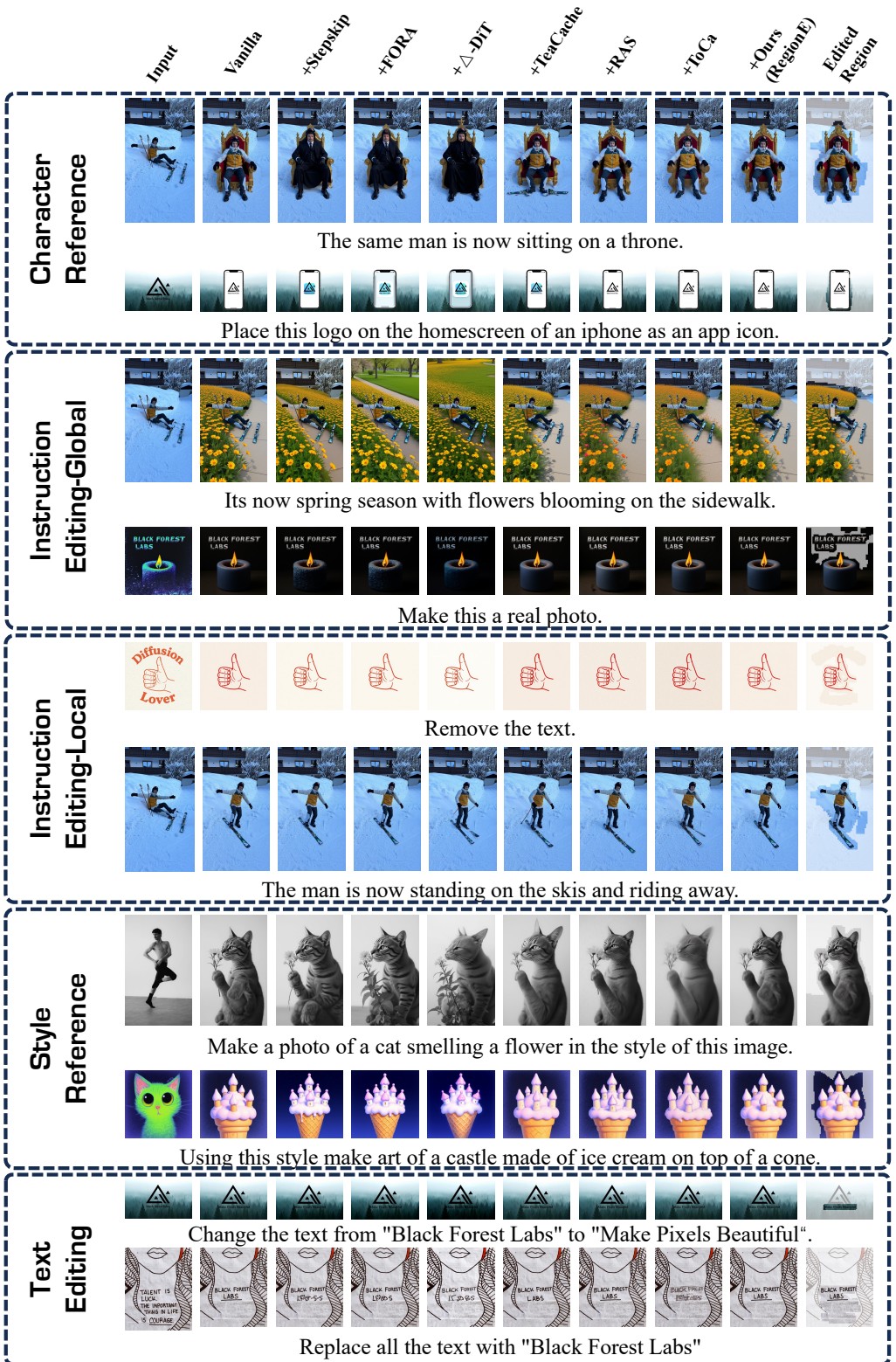

Figure 19: Examples of edited images by RegionE and baseline on FLUX.1 Kontext.

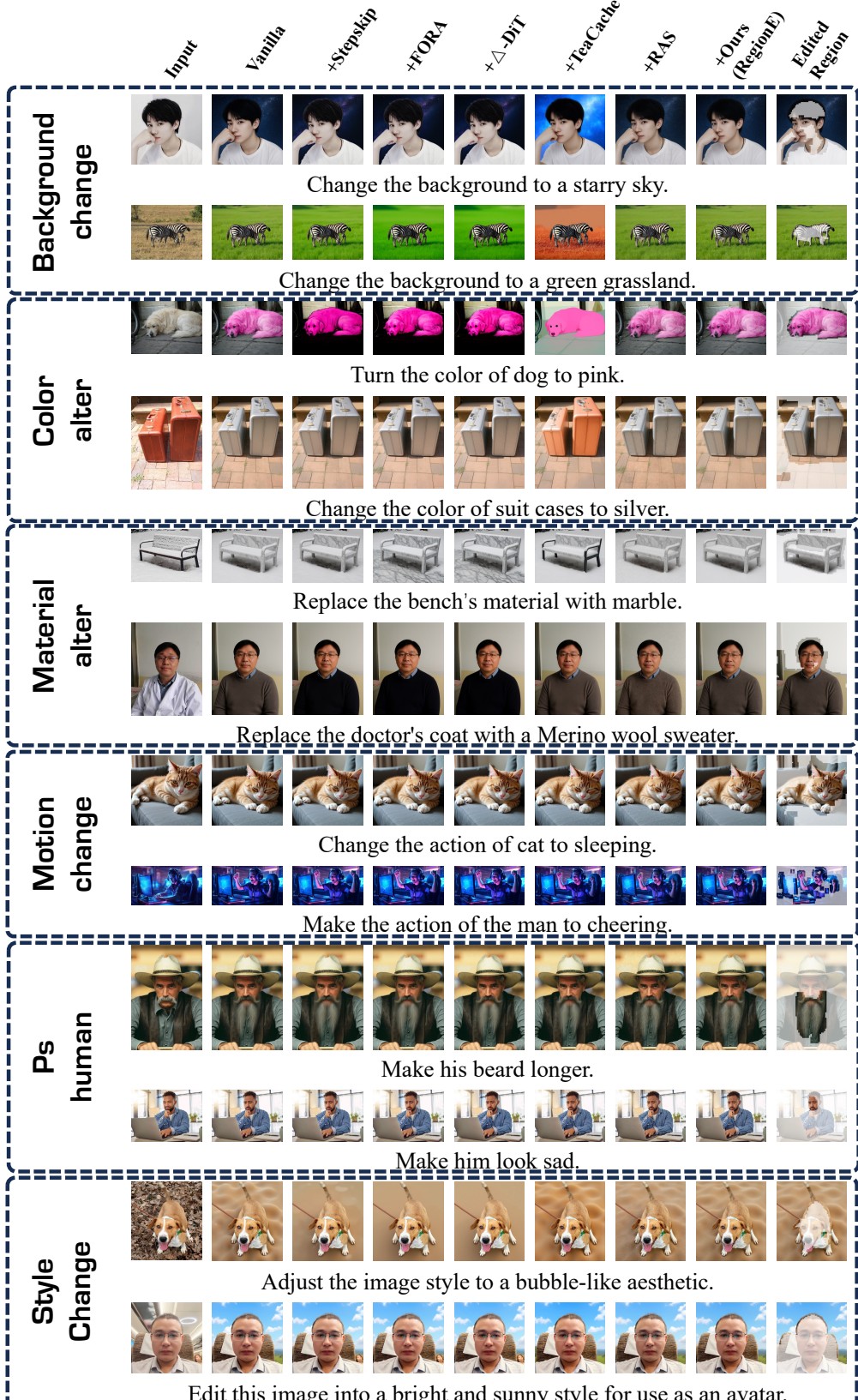

Figure 20: Examples of edited images by RegionE and baseline on Qwen-Image-Edit.

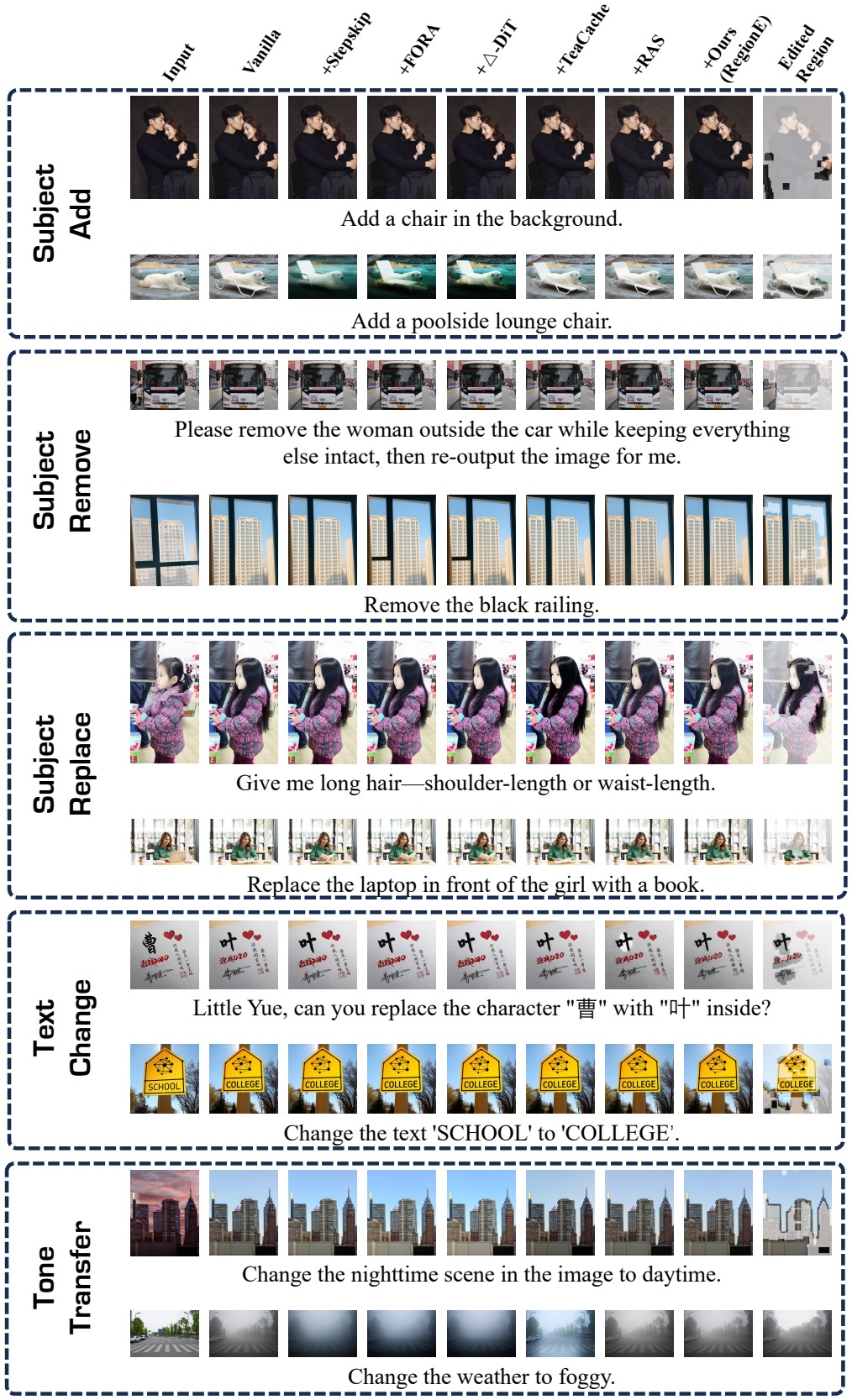

Figure 21: Examples of edited images by RegionE and baseline on Qwen-Image-Edit.

## L   PER-TASK QUANTITATIVE RESULTS IN THE BENCHMARK

In this section, we present the performance of RegionE and the baseline methods on each task in the benchmark. Table 4-Table 14 show the performance on the 11 tasks: motion-change, ps-human, color-alter, material-alter, subject-add, subject-remove, style-change, tone-transfer, subject-replace, text-change, and background-change. Table 15-Table19 show the performance on the five tasks: Character Reference, Style Reference, Text Editing, Instruction Editing-Global, and Instruction Editing-Local.

Table 4: Comparison of RegionE and other baselines on the motion[-change task of GEdit-Bench, evaluated in terms of quality and efficiency.

| Model | Against Vanilla | | | GPT-4o Score | | | Efficiency | |
|---|---|---|---|---|---|---|---|---|
| | PSNR↑ | SSIM↑ | LPIPS↓ | G-SC↑ | G-PQ↑ | G-O↑ | Latency (s)↓ | Speedup↑ |
| **Step1X-Edit** (Liu et al., 2025b) | - | - | - | 4.350 | 7.950 | 4.444 | 27.950 | 1.000 |
| + **Stepskip** | 25.887 | 0.902 | 0.093 | 4.350 | 8.100 | 4.562 | 12.306 | 2.271 |
| + **FORA** (Selvaraju et al., 2024) | 20.935 | 0.818 | 0.189 | 2.175 | 7.575 | 2.385 | 14.339 | 1.949 |
| + **Δ-DiT** (Chen et al.) | 24.549 | 0.876 | 0.121 | 4.350 | 7.975 | 4.445 | 12.730 | 2.196 |
| + **TeaCache** (Liu et al., 2025a) | 26.926 | 0.925 | 0.068 | 4.475 | 8.050 | 4.524 | 11.218 | 2.492 |
| + **RAS** (Liu et al., 2025c) | 25.888 | 0.889 | 0.109 | 4.025 | 7.375 | 4.012 | 15.253 | 1.832 |
| + **ToCa** (Zou et al., 2024a) | 24.428 | 0.843 | 0.165 | 3.775 | 6.975 | 3.578 | 22.225 | 1.258 |
| + **Ours (RegionE)** | 29.633 | 0.937 | 0.053 | 4.625 | 7.775 | 4.763 | 10.739 | 2.603 |
| **Qwen-Image-Edit** (Wu et al., 2025) | - | - | - | 4.850 | 8.550 | 5.112 | 32.140 | 1.000 |
| + **Stepskip** | 27.791 | 0.905 | 0.066 | 4.725 | 8.625 | 5.029 | 17.566 | 1.830 |
| + **FORA** (Selvaraju et al., 2024) | 26.744 | 0.889 | 0.079 | 4.825 | 8.325 | 4.995 | 17.827 | 1.803 |
| + **Δ-DiT** (Chen et al.) | 25.756 | 0.848 | 0.095 | 4.675 | 8.575 | 4.921 | 17.481 | 1.839 |
| + **TeaCache** (Liu et al., 2025a) | 26.776 | 0.911 | 0.070 | 5.025 | 8.500 | 5.251 | 16.389 | 1.961 |
| + **RAS** (Liu et al., 2025c) | 26.585 | 0.882 | 0.096 | 5.000 | 8.625 | 5.262 | 22.300 | 1.441 |
| + **ToCa** (Zou et al., 2024a) | OOM | OOM | OOM | OOM | OOM | OOM | OOM | OOM |
| + **Ours (RegionE)** | 29.416 | 0.932 | 0.057 | 4.825 | 8.550 | 5.164 | 15.695 | 2.048 |

Table 5: Comparison of RegionE and other baselines on the ps-human task of GEdit-Bench, evaluated in terms of quality and efficiency.

| Model | Against Vanilla | | | GPT-4o Score | | | Efficiency | |
|---|---|---|---|---|---|---|---|---|
| | PSNR↑ | SSIM↑ | LPIPS↓ | G-SC↑ | G-PQ↑ | G-O↑ | Latency (s)↓ | Speedup↑ |
| **Step1X-Edit** (Liu et al., 2025b) | - | - | - | 4.614 | 8.086 | 4.649 | 27.927 | 1.000 |
| + **Stepskip** | 29.220 | 0.916 | 0.069 | 4.600 | 8.086 | 4.728 | 12.296 | 2.271 |
| + **FORA** (Selvaraju et al., 2024) | 23.596 | 0.863 | 0.142 | 3.414 | 8.529 | 3.920 | 14.323 | 1.950 |
| + **Δ-DiT** (Chen et al.) | 26.348 | 0.884 | 0.099 | 4.800 | 8.086 | 4.893 | 12.728 | 2.194 |
| + **TeaCache** (Liu et al., 2025a) | 31.428 | 0.942 | 0.047 | 5.114 | 7.929 | 5.191 | 11.208 | 2.492 |
| + **RAS** (Liu et al., 2025c) | 29.077 | 0.921 | 0.072 | 4.400 | 7.886 | 4.486 | 15.237 | 1.833 |
| + **ToCa** (Zou et al., 2024a) | 26.716 | 0.878 | 0.125 | 4.786 | 7.914 | 4.838 | 22.073 | 1.265 |
| + **Ours (RegionE)** | 32.985 | 0.957 | 0.037 | 4.629 | 8.114 | 4.731 | 10.813 | 2.583 |
| **Qwen-Image-Edit** (Wu et al., 2025) | - | - | - | 5.814 | 8.500 | 5.972 | 32.100 | 1.000 |
| + **Stepskip** | 32.080 | 0.936 | 0.040 | 5.757 | 8.414 | 5.904 | 17.553 | 1.829 |
| + **FORA** (Selvaraju et al., 2024) | 30.120 | 0.920 | 0.049 | 5.700 | 8.443 | 5.933 | 17.816 | 1.802 |
| + **Δ-DiT** (Chen et al.) | 28.323 | 0.887 | 0.062 | 5.743 | 8.500 | 5.911 | 17.462 | 1.838 |
| + **TeaCache** (Liu et al., 2025a) | 32.347 | 0.948 | 0.038 | 5.714 | 8.400 | 5.833 | 16.360 | 1.962 |
| + **RAS** (Liu et al., 2025c) | 29.857 | 0.917 | 0.061 | 5.843 | 8.271 | 5.884 | 22.340 | 1.437 |
| + **ToCa** (Zou et al., 2024a) | OOM | OOM | OOM | OOM | OOM | OOM | OOM | OOM |
| + **Ours (RegionE)** | 33.550 | 0.963 | 0.029 | 6.086 | 8.486 | 6.227 | 15.473 | 2.075 |

Table 6: Comparison of RegionE and other baselines on the color-alter task of GEdit-Bench, evaluated in terms of quality and efficiency.

| Model | Against Vanilla | | | GPT-4o Score | | | Efficiency | |
|---|---|---|---|---|---|---|---|---|
| | PSNR↑ | SSIM↑ | LPIPS↓ | G-SC↑ | G-PQ↑ | G-O↑ | Latency (s)↓ | Speedup↑ |
| **Step1X-Edit** (Liu et al., 2025b) | - | - | - | 8.750 | 6.875 | 7.395 | 28.019 | 1.000 |
| **+ Stepskip** | 27.291 | 0.919 | 0.080 | 8.325 | 6.975 | 7.349 | 12.330 | 2.273 |
| **+ FORA** (Selvaraju et al., 2024) | 21.871 | 0.838 | 0.132 | 8.800 | 7.525 | 7.889 | 14.356 | 1.952 |
| **+ $\Delta$-DiT** (Chen et al.) | 24.942 | 0.901 | 0.107 | 8.075 | 6.600 | 6.968 | 12.770 | 2.194 |
| **+ TeaCache** (Liu et al., 2025a) | 28.084 | 0.938 | 0.050 | 8.525 | 6.950 | 7.345 | 11.242 | 2.492 |
| **+ RAS** (Liu et al., 2025c) | 28.800 | 0.909 | 0.069 | 8.700 | 6.925 | 7.432 | 15.274 | 1.834 |
| **+ ToCa** (Zou et al., 2024a) | 25.917 | 0.864 | 0.118 | 8.600 | 6.725 | 7.232 | 21.996 | 1.274 |
| **+ Ours (RegionE)** | 32.739 | 0.956 | 0.032 | 8.850 | 7.250 | 7.747 | 11.188 | 2.504 |
| **Qwen-Image-Edit** (Wu et al., 2025) | inf | 1.000 | 0.000 | 9.250 | 7.525 | 8.170 | 32.082 | 1.000 |
| **+ Stepskip** | 29.795 | 0.896 | 0.064 | 9.050 | 7.450 | 8.084 | 17.527 | 1.830 |
| **+ FORA** (Selvaraju et al., 2024) | 28.035 | 0.879 | 0.078 | 8.875 | 7.350 | 7.872 | 17.795 | 1.803 |
| **+ $\Delta$-DiT** (Chen et al.) | 25.892 | 0.835 | 0.094 | 9.025 | 7.375 | 8.021 | 17.479 | 1.835 |
| **+ TeaCache** (Liu et al., 2025a) | 30.757 | 0.922 | 0.057 | 8.775 | 7.250 | 7.840 | 16.566 | 1.937 |
| **+ RAS** (Liu et al., 2025c) | 29.132 | 0.909 | 0.060 | 9.150 | 7.050 | 7.860 | 22.356 | 1.435 |
| **+ ToCa** (Zou et al., 2024a) | OOM | OOM | OOM | OOM | OOM | OOM | OOM | OOM |
| **+ Ours (RegionE)** | 33.144 | 0.951 | 0.032 | 9.225 | 7.475 | 8.172 | 15.527 | 2.066 |

Table 7: Comparison of RegionE and other baselines on the material-alter task of GEdit-Bench, evaluated in terms of quality and efficiency.

| Model | Against Vanilla | | | GPT-4o Score | | | Efficiency | |
|---|---|---|---|---|---|---|---|---|
| | PSNR↑ | SSIM↑ | LPIPS↓ | G-SC↑ | G-PQ↑ | G-O↑ | Latency (s)↓ | Speedup↑ |
| **Step1X-Edit** (Liu et al., 2025b) | - | - | - | 8.300 | 6.575 | 7.226 | 27.880 | 1.000 |
| **+ Stepskip** | 24.377 | 0.858 | 0.117 | 8.050 | 5.900 | 6.676 | 12.260 | 2.274 |
| **+ FORA** (Selvaraju et al., 2024) | 20.406 | 0.763 | 0.224 | 7.175 | 6.875 | 6.579 | 14.286 | 1.952 |
| **+ $\Delta$-DiT** (Chen et al.) | 21.995 | 0.829 | 0.154 | 8.025 | 5.975 | 6.695 | 12.685 | 2.198 |
| **+ TeaCache** (Liu et al., 2025a) | 25.630 | 0.875 | 0.099 | 8.175 | 6.000 | 6.796 | 11.163 | 2.498 |
| **+ RAS** (Liu et al., 2025c) | 24.302 | 0.844 | 0.141 | 8.275 | 5.700 | 6.633 | 15.202 | 1.834 |
| **+ ToCa** (Zou et al., 2024a) | 22.503 | 0.793 | 0.186 | 7.850 | 5.450 | 6.352 | 22.306 | 1.250 |
| **+ Ours (RegionE)** | 27.248 | 0.897 | 0.080 | 8.475 | 6.200 | 6.997 | 11.251 | 2.478 |
| **Qwen-Image-Edit** (Wu et al., 2025) | - | - | - | 8.725 | 7.150 | 7.629 | 32.156 | 1.000 |
| **+ Stepskip** | 26.300 | 0.870 | 0.093 | 8.650 | 6.875 | 7.557 | 17.578 | 1.829 |
| **+ FORA** (Selvaraju et al., 2024) | 24.699 | 0.841 | 0.116 | 8.525 | 6.675 | 7.389 | 17.839 | 1.803 |
| **+ $\Delta$-DiT** (Chen et al.) | 23.827 | 0.799 | 0.133 | 8.425 | 6.475 | 7.205 | 17.472 | 1.840 |
| **+ TeaCache** (Liu et al., 2025a) | 26.788 | 0.876 | 0.092 | 8.725 | 6.775 | 7.564 | 16.485 | 1.951 |
| **+ RAS** (Liu et al., 2025c) | 26.927 | 0.862 | 0.098 | 8.400 | 6.625 | 7.192 | 22.357 | 1.438 |
| **+ ToCa** (Zou et al., 2024a) | OOM | OOM | OOM | OOM | OOM | OOM | OOM | OOM |
| **+ Ours (RegionE)** | 30.024 | 0.917 | 0.060 | 8.550 | 6.900 | 7.415 | 15.671 | 2.052 |

Table 8: Comparison of RegionE and other baselines on the subject-add task of GEdit-Bench, evaluated in terms of quality and efficiency.

| Model | Against Vanilla | | | GPT-4o Score | | | Efficiency | |
|---|---|---|---|---|---|---|---|---|
| | PSNR↑ | SSIM↑ | LPIPS↓ | G-SC↑ | G-PQ↑ | G-O↑ | Latency (s)↓ | Speedup↑ |
| **Step1X-Edit** (Liu et al., 2025b) | - | - | - | 8.283 | 7.950 | 7.905 | 27.912 | 1.000 |
| **+ Stepskip** | 25.692 | 0.892 | 0.085 | 8.583 | 8.083 | 8.142 | 12.290 | 2.271 |
| **+ FORA** (Selvaraju et al., 2024) | 21.717 | 0.848 | 0.150 | 6.400 | 8.083 | 6.131 | 14.322 | 1.949 |
| **+ $\Delta$-DiT** (Chen et al.) | 24.203 | 0.868 | 0.099 | 8.017 | 8.050 | 7.655 | 12.727 | 2.193 |
| **+ TeaCache** (Liu et al., 2025a) | 26.413 | 0.914 | 0.073 | 8.600 | 8.067 | 8.177 | 11.204 | 2.491 |
| **+ RAS** (Liu et al., 2025c) | 25.008 | 0.880 | 0.101 | 8.100 | 7.517 | 7.532 | 15.232 | 1.832 |
| **+ ToCa** (Zou et al., 2024a) | 23.524 | 0.820 | 0.159 | 7.650 | 6.950 | 6.939 | 22.062 | 1.265 |
| **+ Ours (RegionE)** | 28.514 | 0.923 | 0.058 | 8.383 | 7.950 | 7.858 | 10.528 | 2.651 |
| **Qwen-Image-Edit** (Wu et al., 2025) | - | - | - | 9.117 | 8.017 | 8.381 | 32.081 | 1.000 |
| **+ Stepskip** | 27.666 | 0.890 | 0.092 | 8.767 | 7.950 | 8.146 | 17.532 | 1.830 |
| **+ FORA** (Selvaraju et al., 2024) | 26.871 | 0.879 | 0.093 | 9.017 | 7.933 | 8.313 | 17.810 | 1.801 |
| **+ $\Delta$-DiT** (Chen et al.) | 25.559 | 0.849 | 0.108 | 8.617 | 7.817 | 7.967 | 17.452 | 1.838 |
| **+ TeaCache** (Liu et al., 2025a) | 28.672 | 0.903 | 0.066 | 8.783 | 7.933 | 8.099 | 16.422 | 1.954 |
| **+ RAS** (Liu et al., 2025c) | 27.398 | 0.891 | 0.081 | 9.100 | 7.933 | 8.267 | 22.278 | 1.440 |
| **+ ToCa** (Zou et al., 2024a) | OOM | OOM | OOM | OOM | OOM | OOM | OOM | OOM |
| **+ Ours (RegionE)** | 30.763 | 0.938 | 0.050 | 8.983 | 8.233 | 8.441 | 15.295 | 2.097 |

Table 9: Comparison of RegionE and other baselines on the subject-remove task of GEdit-Bench, evaluated in terms of quality and efficiency.

| Model | Against Vanilla | | | GPT-4o Score | | | Efficiency | |
|---|---|---|---|---|---|---|---|---|
| | PSNR↑ | SSIM↑ | LPIPS↓ | G-SC↑ | G-PQ↑ | G-O↑ | Latency (s)↓ | Speedup↑ |
| **Step1X-Edit** (Liu et al., 2025b) | - | - | - | 7.351 | 7.947 | 6.973 | 27.954 | 1.000 |
| **+ Stepskip** | 33.649 | 0.954 | 0.038 | 7.579 | 7.684 | 6.969 | 12.300 | 2.273 |
| **+ FORA** (Selvaraju et al., 2024) | 30.330 | 0.943 | 0.062 | 5.474 | 7.895 | 5.285 | 14.330 | 1.951 |
| **+ Δ-DiT** (Chen et al.) | 31.847 | 0.948 | 0.047 | 7.930 | 7.684 | 7.319 | 12.724 | 2.197 |
| **+ TeaCache** (Liu et al., 2025a) | 36.735 | 0.973 | 0.024 | 7.281 | 7.737 | 6.841 | 11.213 | 2.493 |
| **+ RAS** (Liu et al., 2025c) | 32.966 | 0.936 | 0.052 | 7.211 | 7.860 | 6.861 | 15.236 | 1.835 |
| **+ ToCa** (Zou et al., 2024a) | 29.806 | 0.894 | 0.095 | 7.175 | 7.088 | 6.481 | 22.378 | 1.249 |
| **+ Ours (RegionE)** | 35.772 | 0.963 | 0.028 | 7.719 | 7.737 | 7.182 | 10.453 | 2.674 |
| **Qwen-Image-Edit** (Wu et al., 2025) | - | - | - | 8.965 | 8.246 | 8.477 | 32.170 | 1.000 |
| **+ Stepskip** | 32.187 | 0.913 | 0.048 | 9.035 | 8.298 | 8.558 | 17.572 | 1.831 |
| **+ FORA** (Selvaraju et al., 2024) | 29.288 | 0.865 | 0.072 | 9.175 | 7.930 | 8.475 | 17.820 | 1.805 |
| **+ Δ-DiT** (Chen et al.) | 27.056 | 0.826 | 0.090 | 8.895 | 7.947 | 8.348 | 17.486 | 1.840 |
| **+ TeaCache** (Liu et al., 2025a) | 31.687 | 0.899 | 0.051 | 8.895 | 8.228 | 8.441 | 16.434 | 1.958 |
| **+ RAS** (Liu et al., 2025c) | 28.440 | 0.876 | 0.080 | 8.842 | 8.035 | 8.371 | 22.331 | 1.441 |
| **+ ToCa** (Zou et al., 2024a) | OOM | OOM | OOM | OOM | OOM | OOM | OOM | OOM |
| **+ Ours (RegionE)** | 32.122 | 0.925 | 0.052 | 9.333 | 8.351 | 8.787 | 15.349 | 2.096 |

Table 10: Comparison of RegionE and other baselines on the style-change task of GEdit-Bench, evaluated in terms of quality and efficiency.

| Model | Against Vanilla | | | GPT-4o Score | | | Efficiency | |
|---|---|---|---|---|---|---|---|---|
| | PSNR↑ | SSIM↑ | LPIPS↓ | G-SC↑ | G-PQ↑ | G-O↑ | Latency (s)↓ | Speedup↑ |
| **Step1X-Edit** (Liu et al., 2025b) | - | - | - | 8.150 | 6.917 | 7.359 | 27.898 | 1.000 |
| **+ Stepskip** | 21.064 | 0.828 | 0.185 | 8.183 | 6.583 | 7.199 | 12.277 | 2.272 |
| **+ FORA** (Selvaraju et al., 2024) | 15.851 | 0.680 | 0.372 | 7.183 | 7.017 | 6.883 | 14.300 | 1.951 |
| **+ Δ-DiT** (Chen et al.) | 18.893 | 0.791 | 0.233 | 8.167 | 6.367 | 7.066 | 12.684 | 2.200 |
| **+ TeaCache** (Liu et al., 2025a) | 21.695 | 0.857 | 0.156 | 8.000 | 6.733 | 7.213 | 11.187 | 2.494 |
| **+ RAS** (Liu et al., 2025c) | 21.355 | 0.814 | 0.193 | 8.217 | 6.400 | 7.108 | 15.217 | 1.833 |
| **+ ToCa** (Zou et al., 2024a) | 19.819 | 0.760 | 0.250 | 8.283 | 6.000 | 6.927 | 22.327 | 1.250 |
| **+ Ours (RegionE)** | 25.449 | 0.900 | 0.102 | 8.267 | 6.617 | 7.251 | 11.797 | 2.365 |
| **Qwen-Image-Edit** (Wu et al., 2025) | - | - | - | 8.267 | 7.133 | 7.526 | 32.115 | 1.000 |
| **+ Stepskip** | 23.954 | 0.805 | 0.139 | 8.067 | 7.083 | 7.355 | 17.560 | 1.829 |
| **+ FORA** (Selvaraju et al., 2024) | 21.784 | 0.745 | 0.185 | 8.017 | 7.100 | 7.385 | 17.807 | 1.804 |
| **+ Δ-DiT** (Chen et al.) | 20.552 | 0.662 | 0.219 | 8.117 | 7.033 | 7.395 | 17.455 | 1.840 |
| **+ TeaCache** (Liu et al., 2025a) | 23.137 | 0.816 | 0.152 | 8.300 | 7.133 | 7.544 | 16.414 | 1.957 |
| **+ RAS** (Liu et al., 2025c) | 24.073 | 0.772 | 0.169 | 7.983 | 7.000 | 7.348 | 22.275 | 1.442 |
| **+ ToCa** (Zou et al., 2024a) | OOM | OOM | OOM | OOM | OOM | OOM | OOM | OOM |
| **+ Ours (RegionE)** | 27.980 | 0.897 | 0.073 | 8.233 | 7.250 | 7.583 | 16.822 | 1.909 |

Table 11: Comparison of RegionE and other baselines on the tone-transfer task of GEdit-Bench, evaluated in terms of quality and efficiency.

| Model | Against Vanilla | | | GPT-4o Score | | | Efficiency | |
|---|---|---|---|---|---|---|---|---|
| | PSNR↑ | SSIM↑ | LPIPS↓ | G-SC↑ | G-PQ↑ | G-O↑ | Latency (s)↓ | Speedup↑ |
| **Step1X-Edit** (Liu et al., 2025b) | - | - | - | 6.950 | 7.325 | 6.679 | 27.874 | 1.000 |
| **+ Stepskip** | 24.744 | 0.899 | 0.122 | 7.200 | 7.325 | 6.917 | 12.260 | 2.274 |
| **+ FORA** (Selvaraju et al., 2024) | 19.078 | 0.786 | 0.251 | 6.900 | 8.125 | 7.088 | 14.293 | 1.950 |
| **+ Δ-DiT** (Chen et al.) | 22.104 | 0.862 | 0.164 | 7.000 | 7.250 | 6.852 | 12.678 | 2.199 |
| **+ TeaCache** (Liu et al., 2025a) | 27.915 | 0.933 | 0.072 | 6.825 | 7.400 | 6.600 | 11.171 | 2.495 |
| **+ RAS** (Liu et al., 2025c) | 26.455 | 0.895 | 0.111 | 7.200 | 7.000 | 6.688 | 15.187 | 1.835 |
| **+ ToCa** (Zou et al., 2024a) | 23.954 | 0.840 | 0.159 | 6.500 | 6.550 | 5.991 | 22.408 | 1.244 |
| **+ Ours (RegionE)** | 30.860 | 0.945 | 0.064 | 6.900 | 7.275 | 6.641 | 11.496 | 2.425 |
| **Qwen-Image-Edit** (Wu et al., 2025) | - | - | - | 8.475 | 8.025 | 8.084 | 32.160 | 1.000 |
| **+ Stepskip** | 29.715 | 0.862 | 0.092 | 8.150 | 8.000 | 7.820 | 17.562 | 1.831 |
| **+ FORA** (Selvaraju et al., 2024) | 27.514 | 0.839 | 0.117 | 8.025 | 7.875 | 7.771 | 17.841 | 1.803 |
| **+ Δ-DiT** (Chen et al.) | 25.471 | 0.792 | 0.139 | 7.950 | 7.725 | 7.592 | 17.462 | 1.842 |
| **+ TeaCache** (Liu et al., 2025a) | 30.064 | 0.910 | 0.061 | 8.375 | 8.125 | 8.033 | 16.381 | 1.963 |
| **+ RAS** (Liu et al., 2025c) | 29.142 | 0.880 | 0.089 | 8.500 | 8.075 | 8.142 | 22.372 | 1.437 |
| **+ ToCa** (Zou et al., 2024a) | OOM | OOM | OOM | OOM | OOM | OOM | OOM | OOM |
| **+ Ours (RegionE)** | 34.051 | 0.948 | 0.034 | 8.450 | 8.275 | 8.199 | 15.851 | 2.029 |

Table 12: Comparison of RegionE and other baselines on the subject-replace task of GEdit-Bench, evaluated in terms of quality and efficiency.

| Model | Against Vanilla | | | GPT-4o Score | | | Efficiency | |
|---|---|---|---|---|---|---|---|---|
| | PSNR↑ | SSIM↑ | LPIPS↓ | G-SC↑ | G-PQ↑ | G-O↑ | Latency (s)↓ | Speedup↑ |
| **Step1X-Edit** (Liu et al., 2025b) | - | - | - | 8.650 | 7.233 | 7.718 | 27.983 | 1.000 |
| + **Stepskip** | 25.233 | 0.875 | 0.111 | 8.683 | 6.867 | 7.548 | 12.325 | 2.270 |
| + **FORA** (Selvaraju et al., 2024) | 20.594 | 0.831 | 0.189 | 5.833 | 6.817 | 5.306 | 14.359 | 1.949 |
| + **Δ-DiT** (Chen et al.) | 22.927 | 0.835 | 0.141 | 8.500 | 6.733 | 7.345 | 12.766 | 2.192 |
| + **TeaCache** (Liu et al., 2025a) | 25.856 | 0.915 | 0.088 | 8.417 | 7.183 | 7.536 | 11.245 | 2.488 |
| + **RAS** (Liu et al., 2025c) | 25.072 | 0.888 | 0.116 | 8.250 | 6.433 | 6.996 | 15.268 | 1.833 |
| + **ToCa** (Zou et al., 2024a) | 23.407 | 0.840 | 0.168 | 8.267 | 6.217 | 6.909 | 22.080 | 1.267 |
| + **Ours (RegionE)** | 28.654 | 0.935 | 0.064 | 8.517 | 7.167 | 7.585 | 10.647 | 2.628 |
| **Qwen-Image-Edit** (Wu et al., 2025) | inf | 1.000 | 0.000 | 8.883 | 7.683 | 8.136 | 32.161 | 1.000 |
| + **Stepskip** | 26.344 | 0.897 | 0.076 | 8.783 | 7.733 | 8.128 | 17.575 | 1.830 |
| + **FORA** (Selvaraju et al., 2024) | 24.578 | 0.864 | 0.104 | 8.600 | 7.550 | 7.930 | 17.836 | 1.803 |
| + **Δ-DiT** (Chen et al.) | 23.745 | 0.829 | 0.120 | 8.450 | 7.267 | 7.687 | 17.496 | 1.838 |
| + **TeaCache** (Liu et al., 2025a) | 25.993 | 0.891 | 0.084 | 8.700 | 7.733 | 8.120 | 16.391 | 1.962 |
| + **RAS** (Liu et al., 2025c) | 25.579 | 0.881 | 0.095 | 8.867 | 7.400 | 7.996 | 22.341 | 1.440 |
| + **ToCa** (Zou et al., 2024a) | OOM | OOM | OOM | OOM | OOM | OOM | OOM | OOM |
| + **Ours (RegionE)** | 29.388 | 0.938 | 0.047 | 8.967 | 7.767 | 8.242 | 15.446 | 2.082 |

Table 13: Comparison of RegionE and other baselines on the text-change task of GEdit-Bench, evaluated in terms of quality and efficiency.

| Model | Against Vanilla | | | GPT-4o Score | | | Efficiency | |
|---|---|---|---|---|---|---|---|---|
| | PSNR↑ | SSIM↑ | LPIPS↓ | G-SC↑ | G-PQ↑ | G-O↑ | Latency (s)↓ | Speedup↑ |
| **Step1X-Edit** (Liu et al., 2025b) | - | - | - | 8.293 | 8.091 | 7.900 | 28.027 | 1.000 |
| + **Stepskip** | 30.069 | 0.955 | 0.032 | 8.222 | 8.192 | 7.951 | 12.331 | 2.273 |
| + **FORA** (Selvaraju et al., 2024) | 26.368 | 0.941 | 0.049 | 7.000 | 7.899 | 6.926 | 14.375 | 1.950 |
| + **Δ-DiT** (Chen et al.) | 28.615 | 0.953 | 0.032 | 8.515 | 8.222 | 8.171 | 12.768 | 2.195 |
| + **TeaCache** (Liu et al., 2025a) | 31.420 | 0.967 | 0.023 | 8.222 | 8.192 | 7.925 | 11.254 | 2.491 |
| + **RAS** (Liu et al., 2025c) | 28.434 | 0.939 | 0.042 | 7.929 | 7.970 | 7.649 | 15.270 | 1.835 |
| + **ToCa** (Zou et al., 2024a) | 26.305 | 0.902 | 0.078 | 7.949 | 7.707 | 7.609 | 21.723 | 1.290 |
| + **Ours (RegionE)** | 32.404 | 0.968 | 0.020 | 8.212 | 8.242 | 8.002 | 10.237 | 2.738 |
| **Qwen-Image-Edit** (Wu et al., 2025) | - | - | - | 9.192 | 8.394 | 8.606 | 32.071 | 1.000 |
| + **Stepskip** | 29.577 | 0.929 | 0.047 | 8.828 | 8.222 | 8.202 | 17.519 | 1.831 |
| + **FORA** (Selvaraju et al., 2024) | 27.408 | 0.909 | 0.061 | 8.818 | 8.303 | 8.192 | 17.790 | 1.803 |
| + **Δ-DiT** (Chen et al.) | 25.837 | 0.881 | 0.072 | 8.879 | 8.333 | 8.259 | 17.432 | 1.840 |
| + **TeaCache** (Liu et al., 2025a) | 29.126 | 0.932 | 0.047 | 8.778 | 8.222 | 8.184 | 16.539 | 1.939 |
| + **RAS** (Liu et al., 2025c) | 26.732 | 0.912 | 0.061 | 8.889 | 8.010 | 8.187 | 22.302 | 1.438 |
| + **ToCa** (Zou et al., 2024a) | OOM | OOM | OOM | OOM | OOM | OOM | OOM | OOM |
| + **Ours (RegionE)** | 31.357 | 0.950 | 0.033 | 8.838 | 8.313 | 8.260 | 14.813 | 2.165 |

Table 14: Comparison of RegionE and other baselines on the background-change task of GEdit-Bench, evaluated in terms of quality and efficiency.

| Model | Against Vanilla | | | GPT-4o Score | | | Efficiency | |
|---|---|---|---|---|---|---|---|---|
| | PSNR↑ | SSIM↑ | LPIPS↓ | G-SC↑ | G-PQ↑ | G-O↑ | Latency (s)↓ | Speedup↑ |
| **Step1X-Edit** (Liu et al., 2025b) | - | - | - | 8.575 | 7.175 | 7.722 | 27.886 | 1.000 |
| + **Stepskip** | 21.011 | 0.812 | 0.218 | 8.625 | 6.975 | 7.635 | 12.267 | 2.273 |
| + **FORA** (Selvaraju et al., 2024) | 15.897 | 0.719 | 0.372 | 6.500 | 7.125 | 6.104 | 14.285 | 1.952 |
| + **Δ-DiT** (Chen et al.) | 18.641 | 0.781 | 0.271 | 8.375 | 6.625 | 7.338 | 12.687 | 2.198 |
| + **TeaCache** (Liu et al., 2025a) | 23.549 | 0.866 | 0.151 | 8.375 | 6.725 | 7.379 | 11.163 | 2.498 |
| + **RAS** (Liu et al., 2025c) | 25.468 | 0.840 | 0.169 | 8.425 | 6.725 | 7.372 | 15.206 | 1.834 |
| + **ToCa** (Zou et al., 2024a) | 22.925 | 0.768 | 0.255 | 8.200 | 6.175 | 6.995 | 22.631 | 1.232 |
| + **Ours (RegionE)** | 29.076 | 0.917 | 0.091 | 8.500 | 7.125 | 7.675 | 11.324 | 2.463 |
| **Qwen-Image-Edit** (Wu et al., 2025) | - | - | - | 9.125 | 8.200 | 8.603 | 32.221 | 1.000 |
| + **Stepskip** | 25.088 | 0.854 | 0.133 | 9.175 | 7.975 | 8.511 | 17.603 | 1.830 |
| + **FORA** (Selvaraju et al., 2024) | 22.473 | 0.802 | 0.184 | 8.775 | 7.875 | 8.252 | 17.815 | 1.809 |
| + **Δ-DiT** (Chen et al.) | 21.263 | 0.750 | 0.210 | 8.825 | 7.850 | 8.281 | 17.552 | 1.836 |
| + **TeaCache** (Liu et al., 2025a) | 24.016 | 0.852 | 0.147 | 8.850 | 7.950 | 8.281 | 16.492 | 1.954 |
| + **RAS** (Liu et al., 2025c) | 26.550 | 0.858 | 0.128 | 9.100 | 7.450 | 8.153 | 22.411 | 1.438 |
| + **ToCa** (Zou et al., 2024a) | OOM | OOM | OOM | OOM | OOM | OOM | OOM | OOM |
| + **Ours (RegionE)** | 30.462 | 0.939 | 0.053 | 9.175 | 8.050 | 8.547 | 16.694 | 1.930 |

Table 15: Comparison of RegionE and other baselines on the Character Reference task of KontextBench, evaluated in terms of quality and efficiency.

| Model | Against Vanilla | | | GPT-4o Score | | | Efficiency | |
|---|---|---|---|---|---|---|---|---|
| | PSNR ↑ | SSIM ↑ | LPIPS ↓ | G-SC ↑ | G-PQ ↑ | G-O ↑ | Latency (s) ↓ | Speedup ↑ |
| **FLUX.1 Kontext** (Labs et al., 2025) | - | - | - | 7.549 | 6.642 | 6.664 | 14.677 | 1.000 |
| **+ Stepskip** | 18.793 | 0.730 | 0.238 | 7.741 | 6.803 | 6.917 | 8.502 | 1.726 |
| **+ FORA** (Selvaraju et al., 2024) | 17.898 | 0.697 | 0.275 | 7.617 | 6.788 | 6.813 | 7.494 | 1.958 |
| **+ Δ-DiT** (Chen et al.) | 15.560 | 0.604 | 0.387 | 7.668 | 6.451 | 6.704 | 6.737 | 2.178 |
| **+ TeaCache** (Liu et al., 2025a) | 20.313 | 0.770 | 0.197 | 7.865 | 6.565 | 6.842 | 6.271 | 2.341 |
| **+ RAS** (Liu et al., 2025c) | 21.320 | 0.752 | 0.214 | 7.632 | 6.352 | 6.657 | 8.211 | 1.788 |
| **+ ToCa** (Zou et al., 2024a) | 19.596 | 0.679 | 0.298 | 7.570 | 6.047 | 6.454 | 11.279 | 1.301 |
| **+ Ours (RegionE)** | 26.980 | 0.880 | 0.086 | 7.637 | 6.611 | 6.715 | 6.406 | 2.291 |

Table 16: Comparison of RegionE and other baselines on the Instruction Editing-Global task of KontextBench, evaluated in terms of quality and efficiency.

| Model | Against Vanilla | | | GPT-4o Score | | | Efficiency | |
|---|---|---|---|---|---|---|---|---|
| | PSNR ↑ | SSIM ↑ | LPIPS ↓ | G-SC ↑ | G-PQ ↑ | G-O ↑ | Latency (s) ↓ | Speedup ↑ |
| **FLUX.1 Kontext** (Labs et al., 2025) | - | - | - | 7.023 | 6.798 | 6.380 | 14.688 | 1.000 |
| **+ Stepskip** | 23.957 | 0.797 | 0.132 | 7.000 | 6.870 | 6.435 | 8.516 | 1.725 |
| **+ FORA** (Selvaraju et al., 2024) | 22.611 | 0.760 | 0.159 | 7.092 | 6.840 | 6.497 | 7.506 | 1.957 |
| **+ Δ-DiT** (Chen et al.) | 18.687 | 0.659 | 0.252 | 7.073 | 6.882 | 6.574 | 6.754 | 2.175 |
| **+ TeaCache** (Liu et al., 2025a) | 27.206 | 0.842 | 0.101 | 7.294 | 6.885 | 6.626 | 6.251 | 2.350 |
| **+ RAS** (Liu et al., 2025c) | 24.845 | 0.778 | 0.157 | 7.302 | 6.866 | 6.668 | 8.221 | 1.787 |
| **+ ToCa** (Zou et al., 2024a) | 23.030 | 0.711 | 0.218 | 7.179 | 6.588 | 6.483 | 11.412 | 1.287 |
| **+ Ours (RegionE)** | 30.403 | 0.886 | 0.071 | 7.126 | 6.943 | 6.572 | 6.379 | 2.303 |

Table 17: Comparison of RegionE and other baselines on the Instruction Editing-Local task of KontextBench, evaluated in terms of quality and efficiency.

| Model | Against Vanilla | | | GPT-4o Score | | | Efficiency | |
|---|---|---|---|---|---|---|---|---|
| | PSNR ↑ | SSIM ↑ | LPIPS ↓ | G-SC ↑ | G-PQ ↑ | G-O ↑ | Latency (s) ↓ | Speedup ↑ |
| **FLUX.1 Kontext** (Labs et al., 2025) | - | - | - | 6.779 | 6.909 | 5.817 | 14.677 | 1.000 |
| **+ Stepskip** | 31.147 | 0.913 | 0.058 | 6.839 | 6.887 | 5.872 | 8.510 | 1.725 |
| **+ FORA** (Selvaraju et al., 2024) | 29.279 | 0.895 | 0.072 | 6.800 | 6.901 | 5.873 | 7.491 | 1.959 |
| **+ Δ-DiT** (Chen et al.) | 23.390 | 0.824 | 0.130 | 6.822 | 6.829 | 5.846 | 6.751 | 2.174 |
| **+ TeaCache** (Liu et al., 2025a) | 33.341 | 0.938 | 0.040 | 6.942 | 6.800 | 5.896 | 6.113 | 2.401 |
| **+ RAS** (Liu et al., 2025c) | 30.088 | 0.907 | 0.063 | 6.945 | 6.921 | 5.972 | 8.219 | 1.786 |
| **+ ToCa** (Zou et al., 2024a) | 26.996 | 0.855 | 0.112 | 6.851 | 6.635 | 5.790 | 11.231 | 1.307 |
| **+ Ours (RegionE)** | 36.334 | 0.959 | 0.025 | 6.889 | 6.880 | 5.917 | 5.799 | 2.531 |

Table 18: Comparison of RegionE and other baselines on the Style Reference task of KontextBench, evaluated in terms of quality and efficiency.

| Model | Against Vanilla | | | GPT-4o Score | | | Efficiency | |
|---|---|---|---|---|---|---|---|---|
| | PSNR ↑ | SSIM ↑ | LPIPS ↓ | G-SC ↑ | G-PQ ↑ | G-O ↑ | Latency (s) ↓ | Speedup ↑ |
| **FLUX.1 Kontext** (Labs et al., 2025) | - | - | - | 6.810 | 6.556 | 6.331 | 14.684 | 1.000 |
| **+ Stepskip** | 18.606 | 0.678 | 0.290 | 6.333 | 6.381 | 5.947 | 8.501 | 1.727 |
| **+ FORA** (Selvaraju et al., 2024) | 17.508 | 0.631 | 0.329 | 6.222 | 6.413 | 5.667 | 7.476 | 1.964 |
| **+ Δ-DiT** (Chen et al.) | 14.639 | 0.525 | 0.450 | 6.397 | 6.873 | 6.108 | 6.731 | 2.182 |
| **+ TeaCache** (Liu et al., 2025a) | 19.781 | 0.712 | 0.264 | 6.444 | 6.286 | 5.832 | 6.261 | 2.345 |
| **+ RAS** (Liu et al., 2025c) | 19.481 | 0.638 | 0.343 | 6.603 | 6.381 | 6.031 | 8.202 | 1.790 |
| **+ ToCa** (Zou et al., 2024a) | 18.245 | 0.553 | 0.439 | 6.000 | 6.175 | 5.668 | 11.480 | 1.279 |
| **+ Ours (RegionE)** | 24.433 | 0.811 | 0.165 | 6.921 | 6.571 | 6.277 | 6.411 | 2.291 |

Table 19: Comparison of RegionE and other baselines on the Text Editing task of KontextBench, evaluated in terms of quality and efficiency.

| Model | Against Vanilla | | | GPT-4o Score | | | Efficiency | |
|---|---|---|---|---|---|---|---|---|
| | PSNR ↑ | SSIM ↑ | LPIPS ↓ | G-SC ↑ | G-PQ ↑ | G-O ↑ | Latency (s) ↓ | Speedup ↑ |
| **FLUX.1 Kontext** (Labs et al., 2025) | - | - | - | 7.826 | 7.913 | 7.295 | 14.697 | 1.000 |
| **+ Stepskip** | 30.950 | 0.943 | 0.033 | 7.717 | 7.750 | 7.142 | 8.535 | 1.722 |
| **+ FORA** (Selvaraju et al., 2024) | 28.976 | 0.915 | 0.044 | 7.696 | 7.543 | 7.067 | 7.520 | 1.954 |
| **+ Δ-DiT** (Chen et al.) | 23.931 | 0.839 | 0.085 | 7.315 | 7.554 | 6.823 | 6.779 | 2.168 |
| **+ TeaCache** (Liu et al., 2025a) | 31.283 | 0.955 | 0.026 | 7.620 | 7.696 | 7.076 | 6.290 | 2.336 |
| **+ RAS** (Liu et al., 2025c) | 27.504 | 0.913 | 0.048 | 7.598 | 7.402 | 6.971 | 8.238 | 1.784 |
| **+ ToCa** (Zou et al., 2024a) | 25.342 | 0.857 | 0.093 | 7.326 | 7.500 | 6.791 | 11.206 | 1.312 |
| **+ Ours (RegionE)** | 34.141 | 0.962 | 0.018 | 7.815 | 7.761 | 7.211 | 5.767 | 2.548 |

## M    STATEMENT ON LLM USAGE

We declare that large language models (LLMs) were used to assist in polishing the writing of this paper. All ideas, methods, and experimental results are original contributions of the authors. The authors take full responsibility for the content of this work.

