# OpenReview forum: "RegionE: Adaptive Region-Aware Generation for Efficient Image Editing"
_ICLR.cc/2026/Conference — ICLR 2026 Poster_

### Official Review · Reviewer_uJ7h · 2025-10-17

**Soundness:** 3
**Presentation:** 3
**Contribution:** 2
**Rating:** 6
**Confidence:** 4

**Summary:**

The paper “RegionE: Adaptive Region-Aware Generation for Efficient Image Editing” proposes a training-free acceleration framework for instruction-based image editing (IIE). Unlike previous diffusion-based editing models that apply uniform denoising to all image regions, RegionE distinguishes between edited and unedited areas to eliminate spatial and temporal redundancy.
The system consists of three key modules:

* Adaptive Region Partition (ARP) — identifies edited and unedited regions based on cosine similarity between estimated and instruction images.

* Region-Instruction KV Cache (RIKVCache) — reuses cached key–value pairs from previous full computations to maintain global context while locally generating edited regions.

* Adaptive Velocity Decay Cache (AVDCache) — models velocity decay across timesteps to accelerate iterative denoising.

**Strengths:**

* The combination of ARP, RIKVCache, and AVDCache addresses both spatial and temporal redundancy coherently and without retraining.

* Evaluations on three major IIE models with consistent metrics and ablations (on cache design and stage structure) convincingly support the claims.

* The paper is technically detailed and easy to follow, with illustrative figures and pseudocode.

**Weaknesses:**

* Pipeline complexity: The full system involves multiple interdependent stages (STS, RAGS, SMS) and caches, making implementation and integration non-trivial.

* Limited discussion of efficiency factors: It remains unclear whether acceleration benefits scale with the size of the edited region—larger edits may reduce efficiency gains.

* Lack of novelty: Although the combination of adaptive partitioning and caching is well-engineered, each component conceptually extends known acceleration ideas (e.g., residual cache, spatial redundancy) rather than introducing a fundamentally new principle.

* Missing user/perceptual study: Although GPT-4o scoring is used, no human user study or subjective evaluation supports perceptual fidelity claims.

* Potential failure cases not analyzed: No discussion of situations where region partitioning might misclassify boundaries or produce artifacts.

* Incomplete citation coverage: Some prior works on region-aware or local editing (e.g., Blended Diffusion, Object-aware Inversion and Reassembly) are not cited or compared.

**Questions:**

* How does the acceleration performance scale with the size or proportion of the edited region? Would larger editing areas significantly reduce the overall efficiency gains?

* To what extent does the proposed method introduce conceptual novelty beyond existing acceleration ideas?

* Have the authors considered conducting a user study or subjective perceptual evaluation to validate the visual quality beyond GPT-4o-based automatic scoring?

* What are the potential failure cases of the proposed adaptive region partition? For example, could boundary misclassification between edited and unedited regions cause visible artifacts?

---

> ### Author Response · Authors · 2025-11-22
> **Response to Reviewer uJ7h (Part 1 / 2)**
>
> Thank you for your efforts to help improve the quality of our manuscript. We have carefully revised the paper according to your suggestions (newly added content is marked in blue). Regarding the weaknesses and questions you raised, we have summarized them into the following six concerns and address them one by one below.
>
> ---
>
> **Concern 1. Pipeline complexity due to multiple interdependent stages (STS, RAGS, SMS) and caches**
>
> RegionE consists of three stages, but from a practical inference perspective, it can be viewed as two processes. The STS and SMS stages follow the same inference logic as the vanilla model, with the only difference being an additional RIKVCache update in STS. The RAGS stage is where the original inference logic is fundamentally modified, converting full-image generation into region generation. Since our method is training-free, it can be applied in a plug-and-play manner, making deployment and practical use highly convenient.
>
>
> ---
>
> **Concern 2. Uncertainty about how acceleration scales with edited region size, with larger edits potentially reducing efficiency gains**
>
> In the revised version of the paper, we added Figure 7. The upper half of this figure shows the speedup achieved by RegionE under different spatial and temporal redundancy settings. As you correctly noted, larger editing regions generally result in lower speedups. However, for FLUX.1 Kontext, Step1X-Edit, and Qwen-Image-Edit, the temporal redundancy is at least 12 steps. Consequently, even for full-image editing, RegionE achieves speedups of 2.12×, 2.18×, and 1.72×, respectively. When only a very small portion of the image is edited, the speedups further increase to at least 2.68×, 2.91×, and 2.01×, respectively.
>
> ---
>
> **Concern 3. Lack of novelty: adaptive partitioning and caching extend existing acceleration ideas rather than introducing fundamentally new principles**
>
> **Novel setting:** Our study is conducted under a new generation of image-editing paradigms, such as Qwen-Image, FLUX.1 Kontext, and Step1X-Edit. These models employ a novel MLLM-based information injection, a DiT architecture tailored for in-context learning, the latest flow-matching training, and a single-stage denoising process. This is fundamentally different from previous paradigms: inversion-based editing requires two stages (noising and denoising), and InstructPix2Pix–style methods generally rely on U-Net architectures, channel-wise conditioning, and traditional DDPM training. To the best of our knowledge, our method is the first acceleration framework specifically designed for this new SOTA editing paradigm characterized by MLLM + DiT, in-context learning, and flow matching.
>
> **Scientific questions:** As stated in the abstract, many current image-editing tasks modify only a small portion of the image. This motivates the development of a method that generates only the local editing region to accelerate the process. We face two key scientific challenges: a). Can the approximate editing region be determined at early denoising steps? b). Can a full-image diffusion model be efficiently adapted for region-specific generation?
>
> **Contributions:**
>
> 1. For the first challenge, we discover a previously unreported property in flow-matching–based editing models: the predicted velocity differs between edited and unedited regions (Fig. 1). Unedited regions can be accurately estimated in a single step, whereas edited regions cannot. Based on this observation, we propose ARP, which identifies approximate editing regions at early steps.
>
> 2. For the second challenge, we propose a local-generation diffusion framework. For the in-context learning DiT, we provide only the prompt and edited-region information for efficiency, while injecting global information into the attention layers via RIKVCache to maintain accuracy. Combining these components converts a full-image diffusion model into a local-generation model that is both efficient and precise. For unedited regions, we use the one-step estimation property identified above.
>
> 3. To further improve acceleration and practicality, we introduce AVDCache to reduce redundancy across timesteps. Appendix B provides a detailed theoretical comparison with existing methods under the latest flow-matching sampling scheme.
>
> In summary, RegionE is the first acceleration method tailored for this new editing paradigm, addressing the novel scientific challenges of early-region identification and local generation, while providing tangible efficiency improvements for practical applications.

---

> ### Author Response · Authors · 2025-11-22
> **Response to Reviewer uJ7h (Part 2 / 2)**
>
> **Concern 4. No human study: GPT-4o scoring is used, but perceptual quality isn’t validated through user or subjective evaluation**
>
> Thank you for your suggestion. Following your advice, we conducted a user study. The experimental setup is as follows: To align with the original experiments, we constructed evaluation sets for RegionE on Step1X-Edit and Qwen-Image-Edit using GEdit-Bench, selecting a total of 11 tasks. For each task, we randomly sampled 5 image–instruction pairs, yielding 55 samples in total. For FLUX.1 Kontext, we used Kontext Bench, selecting 5 tasks with 11 randomly sampled image–instruction pairs per task, also totaling 55 samples. For each evaluation set, we generated edited images using the base model and the base model with RegionE, saving the corresponding results. We then collected votes from 10 participants, who were shown the paired edited images in random order (without knowledge of which method was used). Participants were asked to vote for the image with higher quality and better instruction-following. If the images were comparable, they could choose a neutral option. The final scores were analyzed to compare the two methods. The questionnaire setup is detailed in Appendix I.
>
> The results, shown in Figure 4 of the paper, indicate that users find it difficult to distinguish whether RegionE was used for acceleration. This further validates the high-fidelity performance of RegionE.
>
> ---
>
> **Concern 5. Unanalyzed failure cases: adaptive region partitioning might misclassify boundaries or cause artifacts, such as visible errors at edited–unedited region edges**
>
> **Boundary artifacts:** After reviewing the generated results across the entire evaluation set, we found that RegionE does not produce noticeable boundary artifacts. The main reasons are:
>
> - During region generation, RegionE uses RIKVCache, allowing the local queries to attend to global KV information. Thus, the edited region maintains awareness of the global context, and any boundary artifacts that do appear are extremely minor.
>
> - The final stage of RegionE is the Smooth Stage, which effectively mitigates any residual boundary artifacts between edited and unedited regions.
>
> **Bad cases at high speedups:** At very high acceleration ratios, a few bad cases may appear. After inspecting the full evaluation set, we observed that these are limited to minor generation offsets that do not affect instruction-following quality.
>
> Relevant visualizations and detailed discussions are provided in Appendices E, F, G, and H.
>
> ---
>
> **Concern 6. Incomplete citations of prior region-aware or local editing methods, such as Blended Diffusion and Object-aware Inversion and Reassembly**
>
> Thank you for recommending these works. Below, we provide a brief summary of the two papers and clarify how our method differs from them:
>
> **Blended Diffusion:** Based on natural language prompts and a user-provided local mask, this method proposes a text-driven blended diffusion to edit the masked region. Specifically, during denoising, the sampled output is mixed with the original image’s noisy information to enhance the stability of unedited regions.
>
> **Object-aware Inversion and Reassembly (OIR):** Designed for inversion-based editing, OIR automatically searches for the optimal inversion steps for each editing pair, denoises each region independently, and then seamlessly reassembles them. This approach addresses concept confusion and quality degradation in multi-object editing.
>
> **Differences from RegionE:**
>
> 1. **Editing paradigm:** Blended Diffusion targets an editing paradigm with a given prompt and mask, whereas RegionE operates in a mask-free editing paradigm. OIR is designed for a dual-prompt, two-stage (noising–denoising) inversion-based paradigm, while RegionE targets the new SOTA MLLM + DiT + In-Context Learning + Flow Matching single-stage paradigm.
>
> 2. **Objective:** Blended Diffusion focuses on enhancing unedited regions. OIR aims to reduce inversion steps in inversion-based editing. RegionE focuses on accelerating editing under the new paradigm by converting full-image generation into efficient local-region generation.
>
> 3. **Region identification mechanism:** Blended Diffusion relies on a user-provided mask. OIR identifies individual editing regions using a keyword-driven generator. RegionE leverages a newly discovered property of flow-matching diffusion trajectories (Fig. 1) to distinguish edited from unedited regions.
>
> Although these two works are related to locality and editing, they operate under different paradigms, pursue different objectives, and employ entirely distinct techniques. We have incorporated both papers into the Related Work section and clarified their relationship and differences relative to RegionE.
>
> ---
>
> If you have any further concerns, please feel free to raise them. We would be happy to discuss and exchange ideas further. Once again, we sincerely thank you for your efforts in helping us improve the quality of our paper.

---

### Official Review · Reviewer_Npav · 2025-10-20

**Soundness:** 3
**Presentation:** 3
**Contribution:** 3
**Rating:** 6
**Confidence:** 4

**Summary:**

This paper proposes an acceleration framework for instruction-based image editing tasks. The authors identify two types of redundancy in existing methods: spatial redundancy and temporal redundancy. Spatially, certain image regions remain unchanged before and after editing, and the denoising process for these regions can be significantly accelerated. To address this, the authors design a region partition method to distinguish between fixed regions and editable regions, applying one-step generation to the fixed regions. Temporally, the authors observe strong similarity in velocity between adjacent denoising timesteps, enabling acceleration through a feature cache-based method. Due to the task-specific improvements in the framework's design, it achieves both efficiency and effectiveness enhancements on image editing benchmarks compared with existing acceleration methods.

**Strengths:**

- It effectively tackles both spatial and temporal redundancies. By partitioning images into edited and unedited regions and leveraging temporal similarities between timesteps, it achieves comprehensive speed improvements.
- The proposed framework is specifically tailored for DiT-based instruction image editing. The authors provide some deep insights on DiT cache designs, which may encourage future works.

**Weaknesses:**

- The region partition approach, while highly effective, may not be fundamentally novel. Previous inversion-based image editing methods have been using attention scores to extract masked regions for editing[1][2].
- The temporal acceleration method shares conceptual similarities with existing work in general-purpose diffusion acceleration.

[1]Uniform Attention Maps: Boosting Image Fidelity in Reconstruction and Editing. WACV 2025.
[2]DiffEdit: Diffusion-based semantic image editing with mask guidance. ICLR 2023.

**Questions:**

- Please clarify the strengths of the proposed region partition method compared to previous partition methods in inversion-based editing methods.
- The adaptive velocity decay cache is mainly based on velocity similarities. The results of cache reuse are not explored. For example, at the very early stage of denoising process, the L1 sim / cos sim are low, but the errors introduced by cache reuse may be re-corrected in later denoising process, thus applying cache strategy under low similarities can also be considered.

---

> ### Author Response · Authors · 2025-11-22
> **Response to Reviewer Npav (Part 1 / 2)**
>
> Thank you for your efforts to help improve the quality of our manuscript. We have carefully revised the paper according to your suggestions (newly added content is marked in blue). Regarding the weaknesses and questions you raised, we have summarized them into the following four concerns and address them one by one below.
>
> ---
>
> **Concern 1. Region partition approach effective but potentially not novel, as prior inversion-based methods use attention for masked region editing**
>
> **Different paradigms:** The inversion-based methods you mentioned follow a dual-prompt, two-stage (noising–denoising) editing paradigm. In contrast, our study focuses on the MLLM + DiT architecture under the new in-context learning + flow matching paradigm, which requires only a single-stage denoising process. These are fundamentally different paradigms, and we have clarified this distinction in the Related Work section.
>
> **Regarding the two papers you mentioned (UAM and DiffEdit):**
>
> - **UAM:** Replaces cross-attention maps with uniform attention to improve inversion-based reconstruction. It identifies editing regions by thresholding the cross-attention difference between source and target, mixing uniform attention to enhance editing results.
>
> - **DiffEdit:** Uses the difference between denoised outputs under different textual prompts to obtain a change mask, and then incorporates correction information from the noising process in inversion-based generation to improve semantic image editing.
>
> - **Differences from RegionE:**
>
>   **a. Task:** UAM and DiffEdit target enhancement for inversion-based editing; RegionE focuses on acceleration for in-context learning edit models.
>
>   **b. Region discrimination mechanism:** UAM uses the difference between source cross-attention during noising and target cross-attention during denoising to obtain the editing mask. DiffEdit uses differences between denoised outputs under different prompts. In contrast, RegionE leverages a newly discovered property in flow-matching diffusion models: unedited regions exhibit stable velocities during sampling, enabling early identification of edited vs. unedited regions.
>
> **Underlying scientific questions:** Many image editing tasks modify only a small region, motivating methods that generate just the editing area for efficiency. We address two scientific challenges: (a) Can the editing region be identified early in denoising? (b) Can a full-image diffusion model be efficiently adapted for region-specific generation?
>
> **Contributions:**
>
> 1. For the first question, we discover that in modern flow-matching editing models, the predicted velocity differs between edited and unedited regions (Fig. 1). Unedited regions can be accurately estimated in a single step, whereas edited regions cannot. Based on this observation, we propose ARP, which identifies approximate edited regions at early steps.
>
> 2. For the second question, we propose a region-generation diffusion framework. For the in-context learning DiT architecture, we provide only the prompt and edited-region information for efficiency, while injecting global information via RIKVCache in attention layers to maintain accuracy. Together, these components convert a full-image diffusion model into a region-generation model that is both efficient and precise. For unedited regions, we use the one-step velocity estimation identified above.
>
> 3. To further improve acceleration and practical usability, we introduce AVDCache to reduce redundancy across timesteps. Appendix B provides a detailed theoretical comparison with existing methods under the latest flow-matching sampling scheme.
>
> In summary, RegionE is the first acceleration method tailored for this new editing paradigm, addressing the scientific challenges of early-region identification and local generation, while providing tangible efficiency gains for practical applications.
>
> ---
>
> **Concern 2. Temporal acceleration method conceptually similar to existing general diffusion acceleration techniques**
>
> **High-level design:** The design principle of AVDCache is theoretically discussed in Appendix B. Under current mainstream flow-matching sampling schemes, we derive that a Residual Cache essentially implements velocity decay, with the decay coefficient being a fixed constant determined by the solver. From the velocity-decay perspective, we observe a systematic discrepancy between the solver-based decay and the actual decay during inference. To compensate for this offset, we introduce a timestep-aware correction term.
>
> **Caching mechanism:** An essential component of any caching method is the cache-update strategy. In AVDCache, we are the first to adopt the velocity-decay perspective, using the velocity decay itself as the cache-update rule.
>
> **Application scenarios:** Traditional temporal acceleration methods are typically applied to full-image generation. In RegionE, however, AVDCache is applied to accelerate region generation, which is a fundamentally different use case.

---

> ### Author Response · Authors · 2025-11-22
> **Response to Reviewer Npav (Part 2 / 2)**
>
> **Concern 3. Advantages of the proposed region partition method over previous inversion-based partitioning approaches**
>
> **Paradigm difference:** Inversion-based editing follows a dual-prompt, two-stage (noising–denoising) paradigm with distinct source and target paths. Previous partitioning methods often rely on computing differences between these two paths. However, such approaches are not applicable to our single-stage denoising editing paradigm.
>
> **Attention-free and one-step computation:** Inversion-based partitioning methods typically require either attention information or high-complexity iterative denoising. In contrast, our partitioning method requires only a single-step computation (Fig. 3, lower-left) and does not involve processing complex attention mechanisms.
>
> **Early-stage determination:** RegionE needs to identify edited and unedited regions at early denoising steps. Our proposed ARP leverages the unique sampling properties of flow-matching models to achieve this early-stage discrimination.
>
> ---
>
> **Concern 4. Adaptive velocity-decay caching based on velocity similarity, with unexamined effects of cache reuse at low-similarity early denoising stages**
>
> 1. In RegionE, there are three stages, the first of which is the Stabilization Stage (STS). We do not apply AVDCache in this stage for four main reasons: a). As discussed in Section 4, the input signal-to-noise ratio at this stage is low, and the DiT predictions are inherently unstable, making other acceleration techniques less suitable. b). As you mentioned, the velocity similarity at the beginning is very low. Since this stage is responsible for generating the coarse image outline, applying caching could harm the generation process. c). The stage concludes with the identification of edited and unedited regions, so it is important to avoid introducing errors at this point. d). Prior studies have also suggested that early timesteps should not be subjected to lightweight acceleration, as seen in SVG [1] or ViDiT-Q [2].
>
> 2. For completeness, we also applied AVDCache to the STS. The quantitative results are shown below. While applying AVDCache in the STS yields higher speedups, it causes a noticeable degradation in editing quality: PSNR drops by 1.91, SSIM decreases by 0.013, LPIPS worsens by 0.16, and G-O declines by 0.066. To achieve a better balance between generation quality and efficiency, we therefore choose not to apply AVDCache in the STS.
>
> | Model                 | PSNR↑  | SSIM↑ | LPIPS↓ | G-SC↑ | G-PQ↑ | G-O↑  | Latency (s)↓ | Speedup↑ |
> | --------------------- | ------ | ----- | ------ | ----- | ----- | ----- | ------------ | -------- |
> | **Step1X-Edit**       | -      | -     | -      | 7.479 | 7.466 | 6.906 | 27.945       | 1.000    |
> | *+ Ours wo STS Cache* | 30.520 | 0.939 | 0.054  | 7.552 | 7.405 | 6.948 | 10.865       | 2.572    |
> | *+ Ours w STS Cache*  | 28.610 | 0.926 | 0.070  | 7.455 | 7.395 | 6.882 | 8.583        | 3.256    |
>
> [1] Sparse VideoGen: Accelerating Video Diffusion Transformers with Spatial-Temporal Sparsity, ICML 2025.
>
> [2] ViDiT-Q: Efficient and Accurate Quantization of Diffusion Transformers for Image and Video Generation, ICLR 2025.
>
> ---
>
>
> If you have any further questions or concerns, please feel free to raise them. We would be happy to discuss and exchange ideas further. Once again, we sincerely thank you for your efforts in helping us improve the quality of our manuscript.

---

> > ### Comment · Reviewer_Npav · 2025-11-24
> >
> > Thank you for your responses, which address all my concerns. I have raised my score to 8.

---

> > > ### Author Response · Authors · 2025-11-24
> > >
> > > Thanks for your thoughtful follow-up and for carefully considering our response. We sincerely appreciate your time, your constructive feedback, and your decision to raise the score to 8.

---

### Official Review · Reviewer_nyRe · 2025-11-06

**Soundness:** 3
**Presentation:** 3
**Contribution:** 2
**Rating:** 4
**Confidence:** 4

**Summary:**

This paper introduces RegionE, a training-free framework that accelerates instruction-based image editing by treating edited and unedited regions differently. It detects unedited areas with near-linear trajectories that can be estimated in one step, while edited regions are refined iteratively. Using adaptive region partitioning, cached attention features, and velocity decay reuse, RegionE achieves 2–2.6× faster inference on Step1X-Edit, FLUX.1 Kontext, and Qwen-Image-Edit with minimal quality loss and no retraining required.

**Strengths:**

- Addresses an important and practical issue: the high inference cost of instruction-based image editing.
- Proposes a training-free framework (RegionE) combining spatial and temporal acceleration strategies.
- Demonstrates consistent 2–2.6× speedups across several strong IIE baselines with minimal perceptual degradation.
- Includes solid ablation studies confirming the contributions of key components like RIKVCache and AVDCache.
- The framework is model-agnostic and can be applied to different diffusion-based editors without retraining.

**Weaknesses:**

- The approach is primarily engineering-oriented, integrating existing acceleration techniques (e.g., caching and region partitioning) into a coherent framework focused on practical efficiency.
- Relies heavily on region partition accuracy; there is no ablation or sensitivity analysis of the Adaptive Region Partition (ARP)
- The parameter choices for thresholds $\eta$ and $\delta$ are fixed heuristically, with no explanation or adaptive tuning strategy.
- Limited discussion of computational trade-offs, such as memory footprint or the cost of region masking.
- No discussion of failure cases, such as edits involving multiple or overlapping regions, or complex global changes that may break the region partition.
- In addition to the previous point, the evaluation focuses on overall metrics, lacking analysis of cases where RegionE may underperform (e.g., subtle texture or lighting edits).

(Minor)

- The related work section misses recent region-aware diffusion editing methods such as LIME [1] and Focus on Your Instruction [2], which could better situate RegionE within related spatial editing research.

[1] Simsar, E., Tonioni, A., Xian, Y., Hofmann, T., & Tombari, F. (2025, February). Lime: localized image editing via attention regularization in diffusion models. In WACV'25.

[2] Guo, Q., & Lin, T. (2024). Focus on your instruction: Fine-grained and multi-instruction image editing by attention modulation. In CVPR'24.

**Questions:**

1. How sensitive is RegionE to the partition threshold $\eta$ and decay threshold $\delta$? How did the authors determine these parameters in practice? Could they be learned adaptively from attention maps or reconstruction loss?
2. How does ARP handle ambiguous or overlapping boundaries between edited and unedited regions, especially when the change is gradual or when multiple small edits occur in different parts of the image?
3. Have the authors observed drift or accumulated errors from AVDCache over long denoising sequences, and is the forced full-image update mechanism sufficient to prevent visible artifacts?
4. How does RegionE perform on global or style editing tasks that modify most of the image content? In such cases, does the region partition still provide computational benefits?
5. What is the memory and runtime overhead of the caching mechanisms, and how does RegionE scale with image resolution or batch size compared to standard inference?



(Minor - short discussion)

6. Could the approach extend to video or 3D editing, where temporal/spatial coupling is stronger?

---

> ### Author Response · Authors · 2025-11-22
> **Response to Reviewer nyRe (Part 1 / 4)**
>
> Thank you for your efforts to help improve the quality of our manuscript. We have carefully revised the paper according to your suggestions (newly added content is marked in blue). Regarding the weaknesses and questions you raised, we have summarized them into the following ten concerns and address them one by one below.
>
> ---
>
> **Concern 1. An engineering-driven framework that unifies existing acceleration techniques for practical efficiency**
>
> **New Setting:** Our study is conducted under a new image-editing paradigms, represented by models such as Qwen-Image, FLUX.1 Kontext, and Step1X-Edit. These models adopt a fundamentally new MLLM-based information injection mechanism, a DiT architecture tailored for in-context learning, the latest flow-matching training schemes, and a single-stage denoising process. This is distinct from inversion-based editing paradigms, which require a two-stage noising–denoising procedure, and from InstructPix2Pix-style methods, which predominantly rely on U-Net architectures, channel-wise conditioning, and traditional DDPM training. To the best of our knowledge, our method is the first acceleration framework specifically designed for this new SOTA editing paradigm characterized by MLLM + DiT, in-context learning, and flow matching.
>
> **Scientific Questions:** RegionE is not an engineering-driven proposal; it is motivated by fundamental scientific questions. As stated in the abstract, many modern image-editing tasks require modifying only a small portion of the image. This raises the possibility of accelerating editing by generating only the necessary local regions. To achieve this, we must address two key scientific challenges:
> (1) Can we reliably identify the approximate edited region at very early denoising steps?
> (2) Can we transform a full-image generative diffusion model into an efficient yet accurate region-generation model?
>
> **Contributions:**
>
> (1) For the first scientific question, we discover a previously unreported property of modern flow-matching–based editing models: the predicted velocity differs between edited and unedited regions (see Fig. 1). In particular, unedited regions can be accurately estimated in a single step, whereas edited regions cannot. Leveraging this newly observed phenomenon, we introduce ARP, which allows early-stage discrimination between edited and unedited regions.
>
> (2) For the second scientific question, we propose a region generation diffusion framework. For the new in-context–learning DiT architecture, we achieve efficiency by providing only the prompt and the edited-region information as inputs, while retaining accuracy by injecting global information into the attention layers via RIKVCache. Together, these components convert a full-image diffusion model into a region generation model that is both efficient and precise. For unedited regions, we apply the one-step estimation property identified above to produce the final result directly.
>
> (3) To further enhance the acceleration and practical usability, we introduce AVDCache to reduce redundancy across timesteps. Appendix B provides a detailed theoretical comparison between AVDCache and existing methods under the latest flow-matching sampling procedures.
>
> In summary, RegionE is the first acceleration method tailored for this new editing paradigm, addressing the two core scientific challenges of early-region identification and region generation, while providing substantial real-world efficiency improvements.
>
> ---
>
> **Concern 2. Sensitivity analysis of the Adaptive Region Partition**
>
> In the revised version of the paper, we added a parameter sensitivity analysis of ARP in Figure 6. In ARP, the parameter η controls the separation between edited and unedited regions: a larger η leads to a larger proportion of pixels being classified as edited. As shown in the first column of Figure 6 (i.e., ignoring timestep redundancy), when η decreases, the estimated editing region becomes smaller and less accurate, which in turn degrades generation quality. Specifically, when η decreases from 1 to 0.6, PSNR drops from 32.2 to 30.2, and SSIM decreases from 0.957 to 0.931. However, even when ARP’s region estimation becomes inaccurate, RegionE still maintains a PSNR above 30, which remains strong relative to the baseline methods. Moreover, in our final configuration we set η = 0.88, ensuring that the true edited region is very unlikely to be omitted.

---

> ### Author Response · Authors · 2025-11-22
> **Response to Reviewer nyRe (Part 2 / 4)**
>
> **Concern 3. Heuristically fixed partition/decay thresholds and whether they can be adaptively learned**
>
> RegionE contains two hyperparameters, η and δ, where η controls spatial redundancy and δ controls temporal redundancy. In practice, determining η is straightforward. The guiding principle is to ensure that all truly edited regions are included, even if this introduces a small number of unedited pixels. We select η using a small validation set, ensuring it meets this requirement. After reviewing all editing results across the three base models, we found that a single fixed value of η generalizes well to the entire dataset. As shown in the parameter sensitivity study in Figure 6, further fine-grained adjustments to η bring only marginal improvements. Therefore, once η is chosen for a specific model, it can remain fixed in practical deployment. The choice of δ is then determined based on the target inference-latency requirements. Larger δ values yield higher speedups but also cause moderate quality degradation, whose sensitivity can be observed vertically in Figure 6.
>
> Regarding your question on whether these parameters could be learned: the answer is certainly yes. However, as indicated by the sensitivity analysis in Figure 6, small variations in η lead to very limited gains in generation quality. A more complex learnable-parameter design may yield slight improvements, but the benefit would remain modest. Meanwhile, δ is intentionally exposed as a user-controlled knob, enabling RegionE to operate under varying latency requirements—a design philosophy also adopted in prior work such as TeaCache and Delta-DiT. In terms of performance, RegionE already achieves a high cost-performance ratio, providing 2–2.6× acceleration with no perceptible loss in editing quality. More sophisticated learnable designs could be explored in future work.
>
> ---
>
> **Concern 4. Memory overhead of caching and RegionE’s scaling with resolution or batch size versus standard inference**
>
> 1. **Regarding overhead:** In the revised manuscript, we have added Figure 7, where the bottom section reports GPU memory usage under different RegionE configurations. The memory overhead for FLUX.1 Kontext, Step1x-Edit, and Qwen-Image-Edit is only 6.7%, 10.2%, and 9.2%, respectively, while their achievable upper-bound speedups are 3.15×, 3.39×, and 2.55×.
>
> 2. **Scalability to high-resolution images:** High-resolution image editing exhibits even greater spatial redundancy. Consequently, RegionE yields larger efficiency gains in this setting. We have added relevant discussion and visualization results in Appendix D.
>
> 3. **Scalability with batch size:** Models such as Qwen-Image-Edit use variable-length text tokens and therefore do not natively support batch-mode execution. RegionE inherits this limitation under the default setup. However, with padding or related techniques, batch scalability is possible in principle.
>
> ---
>
> **Concern 5. Failure cases for ARP, including ambiguous/overlapping regions, multiple edits, and complex global changes**
>
> We have added relevant discussion and visualization results for multi-region editing, global editing, and complex editing boundaries in Appendix F and Appendix G.
>
> In fact, dispersed or geometrically complex editing regions do not hinder RegionE’s functionality. This is because RegionE incorporates the RIKVCache mechanism: during attention computation, the local-region queries (Q) still attend to the global key–value (KV) representations. As a result, even highly fragmented or irregular editing regions receive consistent global context, avoiding significant computational bias. The additional visualizations in the appendix further confirm this property.
>
> ---
>
> **Concern 6. The evaluation focuses on overall metrics, lacking analysis of cases where RegionE may underperform (e.g., subtle texture or lighting edits)**
>
> 1. The metrics used in our paper are standard and widely adopted in prior editing and acceleration studies, including Step1X-Edit, Qwen-Image and TeaCache. They therefore provide a scientifically reliable basis for evaluation. Detailed explanations are provided in Section 5.1.
>
> 2. Regarding your concern about texture variation and subtle edits: these aspects are already partially captured by our existing metrics. For example, GPT-PQ evaluates overall image quality, which includes sensitivity to texture fidelity, while GPT-SC measures global instruction-following behavior, which penalizes unreasonable or unintended small-region edits.
>
> 3. Currently, there is very limited research on evaluating local region generation quality, and to the best of our knowledge, no well-established quantitative metric exists for this purpose.
>
> 4. We additionally conducted a user study (Figure 4). The results show that participants found it very difficult to distinguish whether the edited images were accelerated by RegionE. This further validates RegionE’s high-fidelity performance.

---

> ### Author Response · Authors · 2025-11-22
> **Response to Reviewer nyRe (Part 3 / 4)**
>
> **Concern 7. Omission of recent region‑aware diffusion editing methods (e.g. LIME, Focus on Your Instruction) in the related work**
>
> Thank you for recommending these works. Below we provide a brief summary of the two papers and clarify how our method differs from them:
>
> **LIME:** Built on early InstructPix2Pix-style editing, LIME fuses multi-level U-Net features and performs clustering to identify editing regions. It then applies a penalty to irrelevant unedited areas to improve global editing consistency.
>
> **Focus on Your Instruction (FoI):** Also based on early InstructPix2Pix, FoI derives an editing-region mask from cross-attention between image features and instruction keywords. It modulates cross-attention computation and sampling rules for edited vs. unedited regions to enhance multi-instruction editing capabilities.
>
> **Differences from RegionE:**
>
> 1. **Editing paradigm:** LIME and FoI are developed under the early InstructPix2Pix paradigm. In contrast, RegionE targets the new SOTA editing paradigm based on MLLM + DiT + In-Context Learning + Flow Matching, which represents a fundamentally different generation and conditioning mechanism.
>
> 2. **Objective:** LIME and FoI aim to enhance editing-region effectiveness and consistency. RegionE, however, aims to accelerate editing under the new paradigm by converting full-image generation into efficient local generation.
>
> 3. **Region discrimination mechanism:** LIME identifies editing regions via clustering over multi-depth U-Net features. FoI derives region masks using keyword–image cross-attention. Our method instead leverages a newly discovered property (Fig. 1) of flow-matching–based diffusion trajectories: the predicted velocity exhibits consistent differences between edited and unedited regions. RegionE uses this trajectory-based property to distinguish the two regions.
>
> Although these two works are related to locality and editing, they operate under a different paradigm, pursue different goals, and employ entirely different techniques. We have incorporated both papers into the Related Work section and clarified their relationships and distinctions relative to RegionE.
>
> ---
>
> **Concern 8. Potential drift or accumulated errors in AVDCache and effectiveness of full-image updates in preventing artifacts**
>
>
> AVDCache is a temporal caching mechanism: based on the observation that features across neighboring timesteps are highly similar, it accelerates inference by reusing the features from the previous timestep. As with any timestep-level caching, this reuse is an approximation and can in principle introduce small deviations or accumulate errors over time. If the cache were left unchanged for too long, such deviations would inevitably grow. To address this, we propose AVDCache, an adaptive cache-update mechanism that selectively updates the cache at timesteps where the approximation error is minimal. This design prevents the cache from remaining stale for extended periods, ensuring that cumulative deviations remain negligible. The ablation results in Table 2 demonstrate this behavior: AVDCache provides an additional 0.84× speedup, while PSNR, SSIM, and LPIPS degrade by only 0.6, 0.007, and 0.008, respectively. The final PSNR and SSIM still reach 30.52 and 0.94, further confirming that the approximation error introduced by AVDCache is minimal in practice.
>
> ---
>
> **Concern 9. RegionE’s effectiveness on global/style edits and whether region partitioning still offers computational gains**
>
> In the revised version of the paper, we added Figure 7. The upper half of this figure reports the speedup achieved by RegionE under different spatial and temporal redundancy settings. As you correctly noted, larger editing regions naturally lead to lower acceleration. However, for FLUX.1 Kontext, Step1X-Edit, and Qwen-Image-Edit, the temporal redundancy is at least 12 steps. Therefore, even in full-image editing, RegionE still achieves speedups of 2.12×, 2.18×, and 1.72×, respectively. When the edited region is very small, the speedups increase further to at least 2.68×, 2.91×, and 2.01×, respectively.

---

> ### Author Response · Authors · 2025-11-22
> **Response to Reviewer nyRe (Part 4 / 4)**
>
> **Concern 10. Extendable to video or 3D editing**
>
> **Extension to video editing:** Current diffusion-based video generation models share highly similar architectures and training schemes with image generation models. For example, in models such as HunyuanVideo and Wan et al., all video frames are tokenized, concatenated, and fed into a DiT, essentially extending single-image generation to multi-frame generation. If future video editing models adopt the latest in-context learning + flow-matching training paradigm, as seen in Qwen-Image or FLUX.1 Kontext, the only difference would be the increased number of tokens corresponding to multiple frames. Our method would naturally be applicable in such scenarios.
>
> **Extension to 3D editing:** 3D representations can take many representations, such as meshes, point clouds, or voxels, and the associated editing paradigms vary accordingly. If the 3D editing paradigm is based on in-context learning + flow-matching, similar to Qwen-Image, our method can be applied. However, as our expertise in 3D modeling is limited, we are not able to provide a deeper technical discussion on this topic.
>
> ---
>
> If you have any further questions or concerns, please feel free to raise them. We would be happy to discuss and exchange ideas further. Once again, we sincerely thank you for your efforts in helping us improve the quality of our manuscript.

---

> > ### Comment · Reviewer_nyRe · 2025-11-24
> >
> > Thank you for your excellent effort during the rebuttal period and for providing the revised manuscript. My concerns have been clearly and thoroughly addressed. So, I am increasing my rating to 8.

---

> > > ### Author Response · Authors · 2025-11-25
> > >
> > > Thank you for your careful review and constructive feedback, which significantly improved our manuscript. We are pleased that our responses addressed your questions and appreciate your decision to raise the score.

---

### Author Response · Authors · 2025-12-03
**Summary for Area Chairs**

## Note to the Area Chairs

We understand that this year presents unique challenges, and we deeply appreciate the Area Chairs taking on the unprecedented workload of managing the rebuttal and discussion phase under these circumstances. We thank you for your time and effort in evaluating our work.

In the rebuttal, we have carefully addressed all concerns of reviewers point-by-point. As of November 25 (**prior** to the unexpected incident on November 27), we had received positive follow-up feedback from Reviewers nyRe and Npav, both of whom kindly confirmed that our rebuttal had adequately resolved their questions and concerns, and subsequently **adjusted their scores to 8 (i.e., the final score is 886).** For clarity, we also provide a concise summary of the key points from the rebuttal below.

---

## Highlights from Reviews

RegionE establishes a new **training-free acceleration direction** for the **MLLM-injected + DiT + flow-matching single-stage** instruction image editing paradigm. Our findings were acknowledged as **empirically reliable**, revealing a *previously unreported* velocity-trajectory discrepancy between edited vs. unedited regions (Reviewer nyRe). Reviewers highlighted that RegionE **coherently handles spatial & temporal redundancy without retraining** (Reviewer uJ7h), provides **consistent 2–2.6× speedups across strong DiT-based editors** including Step1X-Edit, FLUX.1 Kontext, and Qwen-Image-Edit, and offers **deep insights on DiT cache designs for future research** (Reviewer Npav). Combined with **comprehensive ablations and appended failure-case analysis**, as well as a new **human user study confirming perceptual fidelity**, our method demonstrates practical impact, enabling efficient local region generation while preserving global context through adaptive KV caching.

---

## Addressing Questions and Concerns

Across 3 reviews, reviewers raised **12 distinct questions and concerns** regarding novelty justification, ARP robustness, temporal cache use at early timesteps, edited-region scaling behavior, memory/computation trade-offs, failure cases, and subjective perceptual validation. We resolved **6 of these points via theoretical and paradigm-level clarification** in our rebuttals, addressed the remaining **6 concerns by running new experiments**, and delivered **5 major manuscript enhancements**.

---

## Key Manuscript Changes

All updates are highlighted in blue in the revised PDF. Key additions include:

---

### **1. Sensitivity and Robustness of ARP (Section 5.3 & Appendix F)**

To address reviewer concerns about heuristic thresholds η and partition stability, we provide a detailed sensitivity analysis. We show that decreasing η from 1.0 to 0.6 reduces PSNR from 32.2 to 30.2 and SSIM from 0.957 to 0.931, yet RegionE remains robust with PSNR ≥30, indicating stable spatial partition performance. Our final choice η=0.88 ensures extremely low edited-region miss rate while maintaining generation quality.

---

### **2. Early-Stage Cache Evaluation (Appendix C)**

To test whether temporal caching could be applied at low similarity early denoising steps, we experimented with enabling AVDCache in the STS stage. While this increases speedup from 2.57× to 3.26× on Step1X-Edit, it results in significant perceptual quality degradation: PSNR −1.91, SSIM −0.013, LPIPS +0.016, and G-O −0.066. These results confirm that early timesteps are sensitive and cannot be safely approximated by caching.

---

### **3. Scaling with Edited-Region Proportion (Section 5.3 & Appendix G)**

We evaluate speedup trends from 5% to 100% edited-area proportion. For FLUX.1 Kontext, Step1X-Edit, and Qwen-Image-Edit, the temporal redundancy is at least 12 steps. Therefore, even in full-image editing, RegionE still achieves speedups of 2.12×, 2.18×, and 1.72×, respectively. When the edited region is very small, the speedups increase further to at least 2.68×, 2.91×, and 2.01×, respectively.

---

### **4. Cache Memory and Resolution Scaling Analysis (Section 5.3 & Appendix D)**

We report precise GPU memory overhead from KV and velocity decay caches: 6.7% on FLUX.1 Kontext, 10.2% on Step1X-Edit, and 9.2% on Qwen-Image-Edit. We further demonstrate that high-resolution edits exhibit increased spatial redundancy, leading to proportionally larger speedup gains, confirming favorable scaling behavior.

---

### **5. Human Perceptual User Study (Section 5.2 & Appendix I)**

To validate visual fidelity beyond large-model-based automatic scoring, we conducted a blind A/B user study with 10 participants and 55 total samples sampled from GEdit-Bench and Kontext-Bench. Results show users struggle to distinguish accelerated vs. vanilla outputs, confirming strong perceptual consistency and instruction-following fidelity.

---

We hope these clarifications help you quickly grasp the core discussions with the reviewers and the contributioin of our work. Thank you again for your time and consideration.

---

### Meta-Review · Area_Chair_ZpeT · 2026-01-07

**Summary:**

This paper presents RegionE, a training-free, region-aware acceleration framework for instruction-based image editing under modern DiT + flow-matching paradigms. Reviewers appreciated the practical value, robustness, and empirical thoroughness of RegionE, which achieves 2–2.6× speedups with minimal perceptual degradation across multiple state-of-the-art models.

**Reviewer Concerns:**

Reviewers raised questions about partition sensitivity, memory overhead, generality to global edits, and perceptual fidelity. The authors responded comprehensively with new experiments, theoretical clarifications, scaling analysis, and a human perceptual study. All major concerns were resolved; no critical issues remain outstanding.

**Reviewer Scores:**

nyRe initially rated 4, but after a thorough rebuttal and experimental additions (e.g., partition robustness, failure case analysis), increased to 8.

Npav initially rated 6 and raised concerns on novelty and early-stage reuse but found the explanations and new results convincing, increasing to 8.

uJ7h noted pipeline complexity and underexplored perceptual validation; after new user studies and boundary analysis, the score remains 6, likely unchanged but overall positive.

---

### Decision · Program_Chairs · 2026-01-26

Accept (Poster)